

# Associativity Analysis of SO₂ and NO₂ for Alberta Monitoring Data Using KZ Filtering and Hierarchical Clustering

Joana Soares[1], Paul Andrew Makar[1], Yayne-abeba Aklilu[2], Ayodeji Akingunola[1]

[1]Air Quality Modelling and Integration Section, Air Quality Research Division, Environment and Climate Change, Toronto (ON), M3H 5T4, Canada
[2] Environmental Monitoring and Science Division, Alberta Environment and Parks, Edmonton (AL), postal code, Canada

*Correspondence to*: Joana Soares (joana.soares@canada.ca)

**Abstract.** Associativity analysis is a powerful tool to deal with large-scale datasets by clustering the data on the basis of (dis)similarity, and can be used to assess the efficacy and design of air-quality monitoring networks. We describe here our use of Kolmogorov-Zurbenko filtering and hierarchical clustering of NO₂ and SO₂ passive and continuous monitoring data, to analyse and optimize air quality networks for these species in the province of Alberta, Canada. The methodology applied in this study assesses dissimilarity between monitoring station time series based on two metrics: 1-R, R being the Pearson correlation coefficient, and the Euclidean distance. We have combined the analytic power of hierarchical clustering with the spatial information provided by deterministic air quality model results, using the gridded time series of model output as potential station locations, as a proxy for assessing monitoring network design and for network optimization. We find that both metrics should be used to evaluate the similarity between monitoring time series, since this allows a cross-comparison in terms of temporal variation and magnitude of concentrations to assess station potential redundancy. Here, the relative level of potential redundancy of an existing monitoring location was ranked according to each dissimilarity metric, with sites forming clusters at low values of both 1-R and Euclidean distance being the most redundant. We demonstrate clustering results depend on the air contaminant analyzed, reflecting the difference in the respective emission sources of SO₂ and NO₂ in the region under study. Our work shows that much of the signal identifying the sources of NO₂ and SO₂ emissions resides in shorter time scales (hourly to daily) due to short-term variation of concentrations. However, the methodology nevertheless identifies stations mainly influenced by seasonality, if larger time scales (weekly to monthly) are considered. We have found that data consisting of longer-term averages may lose the short-term variation needed to identify local sources, implying that long-term averaged observations are not suitable for source identification purposes. In addition to averaging time, round-off levels in data reports, and the accuracy of instrumentation were also shown to have a negative influence on the clustering results. We have performed the first dissimilarity analysis based on gridded air-quality model output, and have shown that the methodology is capable of generating maps of sub-regions within which a single station will represent the entire sub-region, to a given level of dissimilarity. Maps of this nature may be combined with other georeferenced data (e.g. road networks, power availability) to assist in monitoring network design. We have also shown that our methodology is capable of identifying different sampling methodologies, as well as identifying outliers (stations' time series which are markedly different from all others in a given dataset).

## 1 Introduction

Air quality monitoring networks are established to obtain objective, reliable and comparable information on the air quality of a specific area, and serve the purposes of supporting measures to reduce impacts on human health and the natural environment, monitoring specific sources, and documenting air quality trends over time. Typically, the site locations of an air quality monitoring network may be determined in response to regulations enforced by government-regulated agencies (e.g. EEA, 1997; US-EPA, 2008), and requires at least some *a priori* knowledge of the expected concentrations and concentration gradients of the pollutants of interest. The latter are highly dependent on the spatial and temporal distribution and magnitude of the emission sources, the physical and chemical properties of the emitted substance, and atmospheric


conditions. The extent to which stations are accessible and the availability of electrical power are additional considerations in monitoring network design. However, recommendations regarding the optimum location and number of monitoring stations may also be achieved by the scientific analysis of existing data. For example, statistical methods making use of existing data have been used to recommend the number and location of monitoring stations required in a network (e.g. Lindley, 1956;

Rhoades, 1973; Husain and Khan, 1983; Caselton and Zidek, 1984). Analytical tools such as Gaussian and Eulerian deterministic dispersion models may also be used to identify possible site locations (e.g. Bauldauf *et al.*, 2002; Mazzeo and Venegas, 2008; Mofarrah and Husain, 2009; Zheng *et al.*, 2011). More recently, the spatial distribution of measured pollutants combined with geostatistical modelling has been used to analyse station data (e.g. Cocheo *et al.*, 2008; Lozano *et al.*, 2009; Ferradás *et al.*, 2010, Zhuang and Liu, 2011).

Cluster analysis is a good example of an analysis approach which assumes, like many statistical methods, that the data analysed contain a certain degree of redundant information, which in turn may be used to describe degrees of similarity or dissimilarity between data records from those stations. Typically applied to large and complex air quality databases to identify spatial patterns based on a metric describing the degree of (dis)similarity between data time series from different stations, cluster analysis (Everitt, *et al.,* 2011) may be used for source identification and network station density

optimization, with a minimum loss of information (Munn, 1981). Hierarchical clustering is a well-established associativity analysis methodology used to determine the inherent or natural groupings of objects, and/or to provide a summarization of data into groups (Johnson and Wicherrn, 2007). The theoretical basis of hierarchical clustering has the advantage of making no assumptions regarding the mutual independence of samples, and does not require examining all clustering possibilities. The similarity among members is established by a distance metric or function, which is used to create a similarity matrix in

which data are cross-compared using the metric. This is followed by operations on the similarity matrix which group data according to their degree of (dis)similarity with respect to that metric. Many studies have aimed to quantify the spatial similarities among monitoring sites in terms of concentration levels and time variation, by applying respectively the Euclidean distance and correlation coefficient as similarity metrics. Studies such as Lavecchia *et al.* (1996), Gabusi and Volta (2005), Gramsh *et al.* (2006), Lu *et al.* (2006) and Giri *et al.* (2007) applied these metrics for analyzing the spatial and

temporal distribution of air contaminants in cities or regions and present possible links between those concentrations with specific sources, topography or meteorological patterns. The majority of these studies focused on ozone ($O_3$) and particulate matter (PM). Saksena *et al.* (2003) applied the methodology to nitrogen dioxide ($NO_2$) and sulfur dioxide ($SO_2$), Ionescu *et al.* (2000) to $NO_2$, Hopke *et al.* (1976), McGregor (1996) to $SO_2$, and Ignaccolo *et al.* (2008) to $PM_{10}$, $NO_2$ and $O_3$. Cluster analysis has also been suggested for monitoring network optimization, including station redundancy analysis in studies such

as Ortuño *et al.* (2005) for CO, Jaimes *et al.*, (2005) and Ibarra-Berastegi *et al.* (2010) for $SO_2$, Omar *et al*. (2005) for aerosol optical properties, Pires *et al.*, (2008) for $O_3$ and PM, and Iizuka *et al.* (2014) for nitrogen oxides (NOx), photochemical oxidant ($O_x$), non-methane hydrocarbons (NMHC), and PM. Solazzo and Galmarini (2015) applied cluster analysis data pre-filtered by iterative moving averages (Kolmogorov-Zurbenko (KZ) filtering, Zurbenko, 1986). Their methodology assessed the similarity of the spectral components of the hourly time series, independent of station location or monitoring technology

employed, without a requirement of prior knowledge of the study area. Their analysis investigated the extent to which concentration time series similarities between the air quality monitoring stations were defined by areas with specific chemical regimes and/or predominant air masses, versus by country borders and/or monitoring network jurisdiction. The latter were identified as resulting from differences in monitoring methodology, reducing comparability of the data across those borders/jurisdictions.

Monitoring of air-quality within and downwind of the oil sands region is a key concern with the provincial and federal governments of Canada. In order to better quantify emissions, downwind chemical transformation, and downwind fate of emitted chemicals from this region, the Governments of Canada and Alberta set–up the Joint Oil Sands Monitoring (JOSM) Plan to "improve, consolidate and integrate the existing disparate monitoring arrangements into a single, transparent



government-led approach with a strong scientific base" (JOSM, 2016). A key part of this overall framework was to develop methodologies to assess the consistency and spatial representativeness of the existing air quality network of the Province of Alberta. The assessment presented here is based on the associativity analysis described in the work of Solazzo and Galmarini (2015) and references therein, and further expands that methodology to focus on monitoring network optimization. The methodology uses the time series of observations at different monitoring stations in Alberta, and analyses this data based on two dissimilarity metrics. Dissimilarity may thus be used to rank stations in terms of potential redundancy, where stations having the lowest levels of dissimilarity may be considered sufficiently similar to be considered potentially redundant.

In addition, we apply the same methodology to time series from a deterministic air-quality forecast model (Global Environmental Multiscale – Modelling Air-quality and Chemistry; GEM-MACH) and assess the extent to which the model output can be used as a potential surrogate for observations in clustering analysis. The combined use of the model and clustering analysis is shown to be a potentially powerful tool for network design, and/or optimization of existent air quality networks.

We introduce the methodology to assess potential redundancy of monitoring stations (Section 2) and describe the observational and model data used to develop the methodology (Section 3). The subsequent sections present the associativity analysis for the continuous monitoring (Section 4), and discuss how the methodology can be used to identify different sampling methodologies (Section 5). We then show how the same methodology may be used with output from an air-quality model. With favourable comparisons to clustering results from air quality monitoring station observations, we show that model output combined with hierarchical clustering provides a new approach for monitoring network design (Section 6). We also discuss potential factors impacting the methodology (Section 7) and our conclusions are drawn in Section 8.

## 2 Monitoring and AQ model data

### 2.1 Study area

Alberta, one of the western provinces of Canada (Figure 1), is the largest producer of conventional crude oil, synthetic crude, and natural gas and gas products in Canada, and is home to one of the world's largest deposits of oil sand (a mixture of clay, sand, water and bitumen) (CAPP, 2016). The monitoring of atmospheric pollutants and the provision of public information on air quality in Alberta is carried out by non-profit organizations called "Airsheds"; these organizations are responsible for air pollution monitoring in specified sub-regions of the province. Figure 1b shows the spatial distribution of these monitoring networks within the province, as well as the largest $NO_2$ and $SO_2$ stack emission sources (National Pollutant Release Inventory (NPRI, 2013). The relative proportion of emissions from different sources depends on the sub-region. For example, in the Athabasca oil sands area (monitored by WBEA stations, red symbols, Figure 1b), $SO_2$ is mainly emitted from stacks (flue-gas desulfurization; "major point sources") and $NO_2$ is emitted from both stacks and off-road vehicle mine-fleets ("area sources"). The 2013 total emissions for Alberta were approximately 681 kt for NOx (NO and $NO_2$) and 311 kt for $SO_2$, respectively.

### 2.2 Monitoring data

We analyse data from both passive and continuous instruments measuring $NO_2$ and $SO_2$ ambient concentrations, the two species that include observations from both measurement methodologies. The nine Airsheds within Alberta are shown in Figure 1b: West Central Airshed Society (WCAS), Wood Buffalo Environmental Association (WBEA), Fort Air Partnership (FAP), Alberta Capital Airshed Alliance (ACAA), Calgary Regional Airshed Zone (CRAZ), Peace Airshed Zone Association (PAZA), Palliser Airshed Society (PAS), Parkland Airshed Management Zone (PAMZ) and Lakeland Industrial Community Association (LICA). Figure 1b colour-codes the sampling site locations by Airshed, with continuous station locations shown as circles and passive stations shown as inverted triangles.



Continuous sampling is typically carried out for regulatory compliance, where high-temporal resolution is required in order to monitor short-term exceedances in highly variable concentrations of pollutants in ambient air. The continuous monitoring principles used to detect and measure $SO_2$ in Alberta are ultraviolet pulsed fluorescence, and chemiluminescence for $NO_2$, and the maximum value for detection limits of the $NO_2$ and $SO_2$ continuous samplers is 1.0 ppbv (AEP, 2014, 2016). In

contrast, passive sampling is carried out in order to determine monthly average ambient air concentrations of atmospheric compounds for determination of long-term trends, assessment of potential ecological exposure risks, and to understand the spatial distribution of the measured pollutant. The majority of the Alberta passive monitors for $NO_2$ and $SO_2$ were developed by Maxxam Analytics Inc. (Tang *et al.*, 1997; Tang *et al.*, 1999; Tang, 2001), with the exception of those employed by PAS (PAS, 2016). The detection limit for 30-day average $NO_2$ and $SO_2$ sampling periods with these samplers is 0.1 ppbv. We

analyse here the data records from 39 continuous and 89 passive $SO_2$ monitoring sites, and 38 continuous and 88 passive $NO_2$ monitoring sites, within the province of Alberta.

Passive sampling techniques have several advantages such as ease of deployment, no power requirements and low maintenance, and have been used as an alternative to continuous monitors for monitoring temporal trends of air pollutants in remote areas (Krupa and Legge, 2000; Cox, 2003; Seethapathy *et al.*, 2008; Bytnerowicz *et al.*, 2010) and evaluation of air

quality of large areas (Gerboles *et al.*, 2006). Their disadvantages are low sensitivity, inability to resolve short duration concentration peaks, and adverse effects of meteorological conditions on reported observations (Tang *et al.* 1997, 1999; Krupa and Legge, 2000; Tang, 2001; Kirby *et al.*, 2001; Partyka *et al.*, 2007; Fraczek *et al.*, 2009; Salem *et al.*, 2009; Zabiegala *et al.*, 2010). Moreover, the passive monitors depend on monthly meteorological information, needed in order to calculate diffusion rates. This information is obtained from the nearest site with meteorological observations, as most Alberta

passive sampling sites do not have collocated meteorological measurements. These constraining factors could influence the sampling and, therefore, the accuracy of the results, causing under- or overestimation of ambient gas concentrations in relation to continuous analysers (Krupa and Legge, 2000).

We first analyse the continuous data, reported as hourly values to AEP for the period from July 2013 through September 2014, in a manner similar to Solazzo and Galmarini (2015), by focusing on the variations associated with different time

scales and the determination of relative redundancy levels for different continuous monitoring stations. The time period for this continuous-only analysis was chosen to overlap with the Environment and Climate Change Canada (ECCC) air quality model simulations (described further in Section 2.3). In a second analysis, continuous and passive observations encompassing the period from February 2009 to December 2015 were analysed together, in an effort to cross-compare the different sampling methodologies. The intent of this second analysis was to determine the extent to which the two

methodologies provide similar results, in addition to determining the relative redundancy levels for the passive monitoring stations. In the second analysis, the continuous data were time-averaged to a similar interval as the passive monitoring data, (the passive data were typically available as monthly or bimonthly averages).

All data were extracted from Alberta and Environment and Parks (AEP) archives (http://airdata.alberta.ca/) and were subjected to additional quality assurance and control (QA/QC) procedures due to the requirement of cluster analysis

methodologies that there are no gaps in the time series of observations. We followed the recommendations of Solazzo and Galmarini (2015), that continuous station data should be rejected if their hourly data records for the analysis period have more than 10% of the total data for the year missing, or contain data gaps of more than 168 consecutive hours in duration Missing data may indicate a calibration period or stations which came on or off line during the analysis period. We also follow their recommendations that data gaps of 1 to 6 hours duration are replaced by the linear interpolation between the

nearest valid data on either side of the gap and, for data gaps of longer duration, the annual average of the non-gap data was used. No substantial difference was found between the resulting cluster analysis by filling the longer gaps with these long-term averages versus using the average of the same number of missing days both before and after the gap.



For the comparison between passive and continuous SO$_2$ and NO$_2$ observations, the hourly continuous station data records were subject to the same station rejection criteria and gap-filling procedures as described above. Passive samplers nominally record either one-month or two-month averages, depending on location. One-month data were averaged to bimonthly data in order to have a consistent time interval for the dataset. When one of the two-monthly values was missing from the original

data, the bimonthly average was treated as missing. Passive stations missing more than 25% of the data over the five year period were rejected from the subsequent analysis. This rejection criterion was less stringent than that applied to continuous data, but was necessary in order to achieve a balance between including monitoring sites with most complete data and attaining good spatial coverage. An inclusion criterion of less than 10% for missing passive data would have reduced the number of SO$_2$ passive sites in the analysis from 52 to 18, and NO$_2$ passive sites from 39 to 18. The missing data were gap-

filled using the averages for the given station for the remainder of the 5 year time period. The gap-filled continuous data for the 5 year period were averaged to the same bimonthly intervals as the passive data. The monitors included in this study are listed in Tables S1, S2, S3 and S4, for the continuous monitoring network analysis for NO$_2$ and SO$_2$ and passive monitoring network analysis for NO$_2$ and SO$_2$, respectively, in Supplement 1.

### 2.3   Modelling output

GEM-MACH (Moran *et al.*, 2010; Makar *et al.*, 2015(a,b), Gong *et al.*, 2015) is an on-line chemical transport model describing several air quality processes, including gas-phase (42 gases), aqueous-phase, and heterogeneous chemistry, and aerosol microphysical processes (9 particle species with a 2-bin sectional representation in the configuration used here). GEM-MACH version 2 simulations were carried out for the period between August 2013 and July 2014, over a domain centred over North America with 10 km grid spacing. The resulting outputs were used as initial and boundary conditions for

a nested set of simulations at 2.5km resolution, for a domain covering the provinces of Alberta and Saskatchewan (Figure 1a). The model was driven by regulatory reported emissions and additional emissions data emissions developed for the model simulations of JOSM (see Zhang *et al.*, 2017, for further details on the model emissions) to better simulate Athabasca oil sand surface mining and processing facilities.

GEM-MACH simulations have been previously evaluated for both NO$_2$ and SO$_2$ concentrations against monitoring network

data, satellite observations and cross-compared to the output of other air quality models, in Im *et al.* (2015), Wang *et al.* (2015), Makar *et al.* (2015a,b), and Moran *et al.* (2016). Further evaluation of GEM-MACH on the high resolution domain used here can be found in Makar *et al.* (2017, this special issue) and Akingunola *et al.* (2017, this special issue).

We use the output from GEM-MACH in two ways – first, hourly 2.5km resolution model results were extracted at monitoring station locations, and cluster analyses for the model and observation data were then compared. This comparison

was carried out in order to evaluate the extent to which the model could act as a proxy for the observations, as well as provide any caveats on the observation analysis associated with time averaging, sampling errors, and accuracy of the observations. In our final analysis, we demonstrate the use of the model as a proxy for monitoring network design, by treating every model grid-cell as if it contained a monitoring station – the clustering analysis of this proxy "data" was then used to define sub-regions within the model domain which could be represented by a single station, for different values of

the clustering metric. We carried out this analysis on a test 36 by 36 cell sub-domain centred on the Athabasca oil sands, but could the methodology could be scaled to larger regions. The result of this final analysis are spatial maps at different levels of a given dissimilarity metric, which may then be used as an aid in determining the locations for observation stations, in an optimized monitoring network.



### 3 Associativity analysis for monitoring data based on dissimilarity

#### 3.1 Separating different time scales using KZ filtering

The KZ filter (Zurbenko, 1986) is a means for removing smaller time scales from a time series, based on an iterative moving average over a specific time window. The combination of the number of times the moving average is applied (m) and the

duration of the averaging window (p) determines the time scales removed from the time series ($KZ_{m,p}$), following the energy characteristics of the filter. Filtering parameters m and p can be derived from the transfer function (see Eskridge *et al.* (1997) and Zurbenko, (1986) for details on the transfer function). The removal of high frequency variations in the data allows different time scales to be isolated and analysed separately. The KZ filter belongs to the class of low-pass filters.

For our analysis, hourly continuous time series data were KZ-filtered to remove short-time-scale variations, resulting in three

additional datasets, which have had filtered out time variations with periods less than a day ($KZ_{17,3}$), a week ($KZ_{95,5}$), and a month ($KZ_{523,3}$). The subsequent analysis may thus examine the effect of removing the signal of the different time scales on the relationships between the stations. The time series resulting from each level of filtering may then be cross-compared, using hierarchical clustering, described in the following section.

In previous work appearing in the literature (Solazzo and Galmarini, 2015), the KZ filter was used in a "band-pass"

configuration. A "band-pass" is the difference between two KZ filters, for two different frequencies, and was used in an attempt to isolate the energy between those two frequencies. However, Hogrefe *et al.* (2000, 2003) indicated that applying the difference in KZ filters for band-pass purposes does not separate the spectral components completely, with the energy spectrum overlapping on between the neighbour components. Rather than each band defining an exclusive set of frequencies, some of the energy from one band could be detected by the neighbouring band. We carried out a detailed analysis of the

band-pass configuration, and confirmed Hogrefe *et al*'s analysis, further finding that this energy "leakage" between bands was sufficient that the frequency bands associated with the shorter time-scales could not be distinguished from each other. However, the KZ filter in its original low-pass form was found to be able to separate the time scales in the test data accurately, simply by choosing m,p coefficients to ensure that all energy was removed below specific frequencies. Subsequent clustering was shown to distinguish the influence of the different time scales, given an appropriate choice of the

filtering parameters m and p. Our detailed analysis of the KZ filter in low-pass and band-pass configurations is described in detail in Supplement 2. Note that the m,p values used in this study were chosen to give an equivalent impact as band-pass filters used in Solazzo and Galmarini (2015).

It should be noted that time *filtering* and time *averaging* do not provide the same information. In the case of low-pass time-filtering, the higher frequency variation above some frequency is *removed* from the time series, while in the case of

averaging, that information is *added* to the average.

#### 3.2 Dissimilarity Analysis using Hierarchical Clustering

"Dissimilarity analysis" encompasses a group of methodologies used to rank datasets based on the extent to which they are different (or *dissimilar*) from each other. Dissimilarity may thus be used to rank stations in terms of potential redundancy such that stations having low levels of dissimilarity may be similar enough to be redundant. One of the most commonly used

methodologies for dissimilarity analysis is hierarchical clustering (Johnson and Wicherrn, 2007).

The first step for hierarchal clustering is to choose a metric to describe how dissimilar the time series are from each other. This metric is then calculated for all possible pairs of the time series comprising the dataset. This initial set of calculations results in a dissimilarity matrix, which may then be used to cluster the data, based on the level of dissimilarity. The pair of time series with the lowest level of dissimilarity is combined and forms the first cluster. The metric of dissimilarity is then

recalculated between the first cluster and the remaining time series, followed by pairing time series and/or clusters with the lowest dissimilarity in the reduced matrix. The number of clusters, which was originally equal to the number of time series in





the original dataset, is thus reduced at each stage of the hierarchical clustering process; the process will be completed when the two last clusters have joined.

In this work, we have used two dissimilarity metrics: (1) 1-R, where R is the Pearson linear correlation coefficient (Solazzo and Gamarini, 2015) and (2) the Euclidean distance (the latter is the square-root of the sum of the squares of the differences

between the two time series' members). The metric based on correlation assesses dissimilarities associated with the changes in concentration as a function of time, while the Euclidian distance metric assesses dissimilarities on the basis of magnitude, over the time period of the analysis. We included the Euclidean distance out of concern that 1-R alone would fail to assess the magnitude differences, which may be more important than correlation, for some monitoring network applications. An extreme example would be two perfectly correlated time series, one of which has an order of magnitude lower average

concentrations than the first; such a comparison could result from two stations positioned at different distances in a line downwind from an emissions source. Using 1-R alone, one of these stations could be considered redundant, despite the information inherent in the lower concentrations associated with increasing distance from the emissions source. For both metrics, the recalculation of the dissimilarity matrix is carried out here with the general averaging method (Næs *et al.*, 2010), as it provides robust and accurate clustering, with a substantial reduction in the processing time required to generate clusters

(Solazzo and Galmarini, 2015).

The level of dissimilarity at which individual station records, and then clusters of records, merge as each new cluster, is called a "node". The order in which station records merge, as well as the level of dissimilarity at which they merge, may be displayed in diagrams known as dendrograms. Dendrograms show the pattern of linkages between nodes as the analysis progressed, with the vertical axis representing the level of dissimilarity, vertical lines representing specific clusters, and

horizontal lines joining the clusters representing the nodes where the clusters are linked. A dendrogram has the appearance of the roots of a tree, with the join between the lowest roots representing the node of the most similar time series, and the trunk of the tree the point at which all data have been joined to clusters. Very similar stations are thus joined at the *bottom* of a dendrogram.

### 3.3   Assessing potential station redundancy

Hierarchical clustering as described above was used to assist in the evaluation of potential monitoring station redundancies, as one of many considerations that could influence decision making on monitoring network design. Having carried out hierarchical clustering using station data, the values of the dissimilarity metric as stations join clusters may be used to define the extent of similarity between stations, as well as a relative ranking of stations based on these similarities. This provides a quick assessment station record similarities and offers insight into how the records are related to each other with respect to

their temporal variations (1-R) and magnitudes (Euclidean distance) throughout the time interval analysed.

An assessment of monitoring record redundancies must be made prudently, the metrics used should be carefully assessed, and the physical distance between the stations and emissions sources should be taken into consideration (see section 7). The inherent limitations of the analysis should also be noted. These include:

1. The ranking of stations is *relative* and specific to a given chemical species, the corresponding set of station time series,

35       and the metric of dissimilarity used in the analysis.

2. Stations excluded because of data incompleteness are not analysed and not evaluated for possible redundancies.

3. The methodology has been applied in the past using observations from existing monitoring stations, in order to analyse the relative dissimilarity between those stations' data records. However, the methodology may *also* be applied to gridded model-generated concentration time series. The latter application provides information on possible new

40       locations for monitoring stations, for a given number of monitoring stations or dissimilarity level (this process is described in more detail in Section 6).



4.  Other considerations may factor strongly into monitoring network decision redundancy; for example, the availability of roads and electrical power, regulatory requirements, cost, etc.

An important corollary to the first point above is that different dissimilarity metrics used in hierarchical clustering may result in different relative rankings of station records. Station records which are highly similar when 1-R is used (this metric is

unitless and zero/unity for the most/least similar time series or clusters), may be highly dissimilar when the Euclidean distance is used (the Euclidean distance will have units of the chemical species being analysed, will be zero for the most similar clusters, but the magnitude of the upper limit of dissimilarity will depend on the specific time series being clustered). "Redundancy" with regards to the metrics examined here is thus *relative* to a given chemical species and dataset used for hierarchical clustering. Therefore, we do not propose specific thresholds of the two metrics for determining redundancy. We

note also that the results of the analyses for two metrics may be combined – station data that are relatively similar under one metric may be examined for their degree of similarity under another metric. The metric levels at which these combinations are examined are themselves also qualitative, but station time series which are highly similar under multiple metrics are in turn a stronger indication of potential redundancy.

Despite the above limitations, the methodology is nevertheless highly useful. In the event of limited available resources for

monitoring, an assessment of relative redundancy, through the use of more than one metric, may aid in decision-making. Aside from implying redundancy between two data records, a high level of similarity may also indicate that a station may provide more information to the network if placed *elsewhere*, as opposed to its current location. In the last part of the analysis (Section 6), we show how the methodology may be extended through the use of air-quality model output to design dissimilarity-optimized air-quality networks.

## 4   Dissimilarity analysis for the continuous monitoring networks in Alberta

### 4.1 Spatial distribution of clusters

The dissimilarity analysis was applied to $NO_2$ and $SO_2$ observational time series data for all the stations complying with the QA/QC criteria described in Section 2. The dendrograms resulting from the analysis are provided in Supplement 1.

The hierarchical clustering results for $NO_2$ using 1-R as the dissimilarity metric are depicted in Figure S1. This $NO_2$

dendrogram shows frequent clustering between stations within the same Airshed (if represented by more than a single station) or Airsheds that are in relatively close physical proximity, such as Airsheds ACAA and FAP (see Figure 1b). A horizontal line cutting across a dendrogram such as Figure S1 may be used to define the station records that are part of a cluster at a given level of the dissimilarity metric, and these may be plotted spatially: Figure 2 shows the spatial distribution of the clusters of $NO_2$ continuous monitors at three levels of the 1-R dissimilarity metric: 0.75 (Figure 2a), 0.65 (Figure 2b)

and 0.55 (Figure 2c). The results show that stations tend to cluster over successively smaller areas as the level of dissimilarity decreases (the three clusters of Figure 2a as dissimilarity decreases become eleven clusters by Figure 2c). The clustering at high dissimilarity levels (aka low correlation coefficients) also allows anomalous groupings of stations. For example, cluster 1 in Figure 2a includes both WBEA stations at the upper right of the panel, one WCAS and one PAMZ station, despite the latter two sampling air in other parts of the province and subject to different sources. This tendency is

reduced at lower levels of dissimilarity, where stations influenced by similar sources tend to cluster. For example, in Figure 2c, cluster 8 includes all the stations in a highly urbanized area (Edmonton, capital city of the province) and cluster 11 is a station located at a relatively high elevation upwind of most emission sources. Overall, the methodology shows the ability to group together monitoring station locations which might be expected to be influenced by similar sources of emissions.

We next examine how the time scales inherent in the data may affect similarities. Figure 3 shows the clustering of stations

which occurs at a 1-R dissimilarity level of 0.55 after time scales less than daily (Figure 3a, dendrogram in Figure S2), weekly (Figure 3b, dendrogram in Figure S3) and monthly (Figure 3c, dendrogram in Figure S4) are removed. Four clusters



are shown on the first panel, three on the second, and two on the third. Comparing back to Figure 2c with the original hourly data, this shows that much of the "signal" in 1-R contributing to the eleven clusters in Figure 2c is contained within the shorter time scales, of less than a day, and are relatively similar at longer time scales. Moreover, correlation levels between stations increase as KZ filtering is applied and shorter time variability is removed. All of this evidence indicates that much of

the variation in $NO_2$ in the region takes place on relatively short time scales and is due to local sources. The analysis also indicates that some stations are more influenced by seasonality than others, e.g., the high altitude, largely upwind site of cluster 2 in Figure 3c remains separate from the other stations even when time scales of less than a month are removed from the analysis.

The dissimilarity analysis for $SO_2$ produced different results from that for $NO_2$. Figure 4 shows the spatial distribution of the

clusters of $SO_2$ continuous monitors with the 1-R dissimilarity metric (the dendrogram resulting from the hierarchical clustering appears in Figure S5), and may be compared to Figure 2. For a given level of 1-R, there are more $SO_2$ clusters than $NO_2$ clusters. The observations of $SO_2$, despite being largely collocated with the observations of $NO_2$, are nevertheless more dissimilar than the observations of $NO_2$. Even at higher levels of dissimilarity (compare Figure 2a and Figure 4a), there are more $SO_2$ clusters, indicating a greater degree of local variability in the $SO_2$ data, which drives correlation coefficients

lower and dissimilarity levels for the 1-R metric higher. This greater degree of dissimilarity for $SO_2$ is due to the nature of the $SO_2$ emissions, i.e., almost exclusively from industrial "point" sources in the region under study, whereas $NO_2$ concentrations are also influenced by more broadly geographically dispersed "area" sources of emissions including mobile on and off-road vehicles. The dispersion of $SO_2$ from the former source type is thus more dependent on very local meteorological conditions governing the rise of buoyant plumes from stacks than are the emissions from area sources. The

direction and concentration of the rising and dispersing $SO_2$ plumes is thus more highly variable in time, compared to the area-source dominated emissions of NO, which are chemically transformed rapidly to $NO_2$. Concentrations from the same $SO_2$ source may therefore not correlate to the same degree between different downwind stations as $NO_2$. This contributes to the lesser degree of similarity between the $SO_2$ station data even when monthly and shorter time scales are removed (the $SO_2$ dendrograms with the removal of time scales less than daily, weekly and monthly appear in Figure S6, Figure S7, and Figure

S8, respectively).

The Euclidean distance dendrograms for both $NO_2$ (Figure S9) and $SO_2$ (Figure S10) do not show the same distinctive clustering within Airshed as can be seen with the 1-R metric. This might be expected, as Euclidean distance between two time series may result from a single instance in which the hourly concentration records of the two stations differ substantially or several hours in which the concentration differences are smaller. Stations located sufficiently far apart that they monitor

different sources of pollutants may thus have similar Euclidean distances if their average concentration magnitude is similar. The analysis also indicates that Euclidean distances become more similar in magnitude, and that these magnitudes decrease, as increasingly larger time scales are filtered, across all of Alberta (Figure S9 for $NO_2$ and Figure S10 for $SO_2$). That is, concentration magnitudes recorded at the different stations approach each other as the shorter duration time variations are removed. At these time scales, the magnitude of both species is driven by low concentration levels of long-term duration and

larger spatial extent. This is particularly true for $SO_2$ monitors that typically measure low concentration (background levels) interspersed with infrequent short-term high concentrations (surface fumigation events of buoyant plumes). However, within an Airshed affected by a common set of emissions sources, Euclidean distance will nevertheless be useful, by identifying the presence of high concentration gradients, as will be shown in the next section.

### 4.2 Ranking of stations by dissimilarity

Previous work appearing in the literature (Solazzo and Gamarini, 2015) was motivated by the aims of evaluation and pre-screening of monitoring data, for the purpose of the evaluation and development of regional-scale air pollution models. Their focus was on observations of ozone which, in the troposphere, is a secondary pollutant resulting from gas-phase reactions



and broader-scale chemistry and transport. They consequently focused on the different time scales associated with KZ filtering. Here, however, we have shown that for primary pollutants such as $SO_2$ and "secondary" pollutants such as $NO_2$ which are nevertheless very rapidly (on time scales of less than 5 minutes) produced from their primary precursors, much of the signal driving similarity resides at shorter time scales. Consequently, our ranking of continuous monitoring stations in

this section is based solely on the original hourly observation data, as opposed to KZ filtered observations.

The cluster analysis results for hourly time series were ranked from highest to lowest values of 1-R and Euclidean distance resulting from clustering of continuous monitoring station data. Stations clustering at high levels of 1-R and Euclidean distances are significantly different in time variation and concentration magnitudes, respectively. Conversely, stations at the bottom of the ranking are the most similar. The latter stations could be, therefore, considered potentially redundant. Our

rankings are based on the dissimilarity level at which a given station joins another station as a new cluster, or when a given station joins a pre-existing cluster. If the latter were to occur at a sufficiently low level of dissimilarity, either the new station or the pre-existing cluster might be considered potentially redundant. The uppermost and lowermost ranked stations for $NO_2$ and $SO_2$ are shown in Tables 1 and 2, respectively. The corresponding full ranking for the full list of stations is show in Tables S5 and S6.

The tabulated values indicate clear differences between the two compounds. The stations measuring $NO_2$ cluster with each other at substantially lower 1-R levels (that is, they correlate at substantially higher values of R) than do the stations measuring $SO_2$. In one extreme case, the records of one $SO_2$ station, Redwater Industrial, *anti*-correlate with the records of other stations, indicating that the $SO_2$ time series at that location is substantially different from those of the remaining stations. However, the $NO_2$ Euclidean distance metric cluster values tend to form at higher levels than their $SO_2$ counterparts,

with the exception of Redwater Industrial, indicating that despite their higher correlations, the $NO_2$ stations may have larger differences in concentration magnitudes relative to $SO_2$. We note that the Euclidean distance between $SO_2$ station observations is, in many cases, relatively low (e.g., 24 ppbv for 8760 hourly values summed), and likely indicates stations which rarely record $SO_2$ concentrations above background levels and hence have relatively "similar" Euclidean distances due to similarly low concentration records for much of the recorded time series. Another interesting difference between the two

atmospheric compounds is that the relative ranking by dissimilarity is closer to being the same for the two metrics for $SO_2$, than for $NO_2$.

Two different dissimilarity metrics thus result in different relative rankings the two chemical species, so the results must be interpreted with care. For example, the stations Fort McKay South and Fort McKay Bertha Ganter have the highest correlation for $SO_2$ (R=0.81) but their Euclidean distance is 177 ppbv, and a similar disparity between 1-R and Euclidean

distance rankings for these stations may be seen in their values of the corresponding $NO_2$ metrics (R=0.84 and Euclidean distance of 411 ppbv). These stations are 4 km apart; the high correlation coefficients indicate that they may measure similar events, but the high Euclidean distances indicate that the magnitude of the events observed likely vary considerably despite the small separation distance. That is, substantial gradients in concentration may exist between the two stations at any given time. We note again here that *low* values of the dissimilarity metrics indicate a greater level of potential redundancy with

respect to the rest of the stations – a *high* value of the Euclidean distance between two station records, or between a station record and a cluster, indicates that they are very *dis*similar, and hence *less* potentially redundant. A second example is the pair of stations measuring $NO_2$ with the lowest 1-R, Ross Creek and Fort Saskatchewan: these stations' data records are highly similar with respect to 1-R, that is, they are highly correlated, but the Euclidean distance between the two is 400 ppbv, despite the stations being separated in distance by only 2.6 km. Again, the gradients in concentration between closely placed

stations can be substantial. The intended purpose of the monitoring at such locations is key in assessing their level of potential redundancy. For example, if the aim of monitoring is to provide short-term exposure data for human health impacts, then these large Euclidean distances (despite the high correlations) indicate the presence of large gradients in concentration, and hence such station pairs should be considered less redundant. The combination of the metrics is thus



shown to be important in network assessment – the addition of the Eulerian distance metric provides a broader context for station-ranking than the use of 1-R alone.

## 5  Hierarchical clustering to cross-compare methodologies and technologies

Solazzo and Galmarini (2015) noted that clustering analysis can be used to determine the extent to which the different

monitoring methodologies are comparable. Thus if different methodologies do not provide equivalent data, the clusters generated will be split according to methodology, rather than being associated with local chemical and meteorological conditions. The combination of both methodologies in a single clustering analysis here thus has two purposes – exploring the relative dissimilarities between the station records, and the extent to which the two methodologies examined here (passive and continuous monitors) provide similar data.

The hierarchical clustering methodology was applied to the five year bimonthly averaged time series sampled by continuous and passive monitors (we leave out the *a priori* KZ filtering step as the data in this case are already long-term averages). The dendrograms resulting from the clustering analysis are shown in Figure S11 for $NO_2$ and Figure S12 for $SO_2$. The spatial distributions for the station clusters for the 1-R dissimilarity metric will be the focus here.

The spatial distributions of the $NO_2$ clusters at dissimilarity levels of 1-R=0.55 and 0.5 are shown in Figure 5a and 5b,

respectively, with the locations of continuous monitors plotted as inverted triangles and passive monitors as circles. At correlation level R=0.45 (Figure 5a) there is a clear distinction between passive and continuous monitors, all the continuous monitors belong to cluster 1, independent of their spatial location. A large number of the passive monitors also fall within this cluster; however, when a slight increase in correlation is applied (Figure 5, R=0.5), the clustering pattern changes significantly – most of the continuous monitors remain within the same cluster, but the passive monitors form separate

clusters. Two WCAS continuous monitors separate and form a separate cluster at dissimilarity level 0.5 (Figure 5b). Figure 5 also shows several cases of *collocated* continuous and passive monitors which do not fall within the same cluster for correlation levels of 0.5 or higher. The analysis shows that as higher levels of correlation are required, the continuous and passive monitors for $NO_2$ do not cluster together despite close physical proximity or even collocation. Some of the passive monitor clusters at R = 0.5 (Figure 5b) appear anomalous; for example, cluster 3 (red) includes stations in LICA and WBEA,

despite these airsheds being separated by a distance of several hundred kilometres. As the level of dissimilarity is decreased from 0.55 to 0.5, the biggest difference in clustering patters is seen for WBEA monitors, in the upper right of the panels of Figure 5, as passive and continuous monitors located closer to the oil sands facilities are fall within cluster 1, while some of the passive monitors farther from the oil sands facilities fall within cluster 3. For levels of correlation above 0.5, the clustering between stations monitoring similar source areas is rare, independent of the Airshed (see dendrogram in Figure

S8).

Figure 6 depicts the clustering results for $SO_2$ based on the 1-R metric for dissimilarity levels 0.75 (Figure 8a) and 0.65 (Figure 8b). Higher dissimilarity levels were used as examples for the generation of spatial distributions, than for $NO_2$ in this Figure. The highly variable nature of the $SO_2$ concentrations, as a result of their origin in stack emissions, results in a greater degree of variability inherent in the collected data, as described earlier (at lower dissimilarity levels, the number of clusters

increases markedly). Comparing Figure S9 and Figure 6, most of WBEA passive and continuous monitors in the north-east of the region form a common cluster at R=0.25 (Figure 6a, cluster 11, red). However, at this low correlation level, a common cluster connects sites in LICA, FAP, WBEA and PAZA Airsheds, despite these sites being widely separated in space and influenced by different local sources of $SO_2$ (cluster 12, green, Figure 6a). At the slightly higher correlation level of R=0.35 (Figure 6b), the clustering across airsheds has been reduced, though LICA and FAP still share a common cluster (number 4,

light blue). Again, the most direct interpretation of the differences between the $SO_2$ and $NO_2$ results for the 1-R metric analysis, when passive and continuous monitors are clustered together, is that the data time series records for $SO_2$ are more



highly variable than for NO$_2$. If 1-R similarity is used for assessing potential station redundancies, then there is a lesser overall degree of potential redundancy in the SO$_2$ data, due to its greater degree of variability. However, the cause of that variability should also be considered. For example, we note again that some of the collocated passive and continuous monitors for SO$_2$ do not fall within the same cluster at lower 1-R values (these are shown as different colours in overlapping inverted triangles and circles in Figure 6b). This indicates that at least some of the variability may reside in the measurement methodologies employed.

In their analysis of European ozone monitoring networks, Solazzo and Galmarini (2015) found similar patterns between different European nations, noting that the differences likely related to different sampling methodologies, instrument sensitivities, and data acquisition protocols not being harmonised between the countries. The same seems to be true for the Alberta passive and continuous monitoring stations, as the 1-R cluster analysis shows that the continuous stations are more similar to each other within and across Airsheds, than they are to the passive stations within the same Airshed, or located nearby. Collocated continuous and passive stations do not always show high levels of similarity, which would be expected, had they reported the same concentrations. We analysed WBEA data alone using the 1-R metric (dendrogram in Figure S13), and found that most of the continuous monitors formed a separate cluster from the passive monitors at relatively high levels of the 1-R metric, indicating that the two sources of data are providing fundamentally different records. Collocated passive and continuous monitors also tended to have high levels of the Euclidean distance (not shown). Thus, at least some of the variability noted with these datasets seems to lie with the overall sampling methodology, and related confounding factors, discussed further in Section 7.

There have been several studies comparing passive and continuous analysers in Alberta (WBK, 2007; Hsu *et al.*, 2010; Pippus, 2012; Bari *et al.*, 2015). Bari *et al.* (2015), the study with the highest number of samples, cautioned that direct comparisons between NO$_2$ and SO$_2$ continuous and passive methods may be hampered by lower field accuracy in the passive methodology. Several studies show that passive samplers overestimate SO$_2$ ambient concentrations and underestimate NO$_2$, relative to continuous monitors. For example, the Bari *et al.* (2015) study showed that the median values for the absolute difference between the collocated passive and continuous monitors for NO$_2$ is 1.5 ppbv and 0.2 ppbv for SO$_2$. The same study assessed the relationship between passive and continuous measurements by regression analysis, concluding that the agreement between the different types of monitors is moderate, with the coefficient of determination being 0.42 and 0.40 for NO$_2$ and SO$_2$ respectively. We note that these previous comparisons were done for urban sites only; in this study we have carried out cluster analysis including passive and continuous monitoring data for rural, urban, and industrial sites outside of urban regions.

## 6    Model information as a potential surrogate for observations: optimized monitoring network design

Air-quality models such as GEM-MACH provide gridded time series concentrations of atmospheric pollutants and related chemicals at a common time interval, as a standard output. These are compared to observations in order to evaluate the model's performance (cf. Makar *et al.*, 2017; Akingunola *et al.*, 2017, Stroud *et al.*, 2017 for traditional evaluations using the model output used herein). We introduce here for the first time the concept of the use of these time series of air-quality model output, combined with hierarchical clustering analysis, as a surrogate for station data, for the purposes of monitoring network analysis and design. Two possible approaches can be taken. First, the model output at the model grid-squares containing existing monitoring stations may be analysed, in order to determine the extent to which the clustering analysis of model output mimics the clustering analysis of the corresponding observational data. Aside from presenting a new means by which the model output can be evaluated, this approach also can highlight possible causes for the observation data clustering results. The second approach is to use the gridded model output as a surrogate for a dense monitoring network (one "station" at every model grid-square center). The outcome of this second approach is a set of gridded maps – similar to the sparsely





distributed observation location maps shown in the figures above, these show the clustering of *potential* stations. However, the cluster maps resulting from the use of the dense "network" of model grid-squares, defines more precisely a set of *regions* within each of which a single station may represent that larger region, for the value of the dissimilarity metric chosen. We investigate this second approach from the standpoint of monitoring network design. Note that, in the work above, we have

attempted to show how hierarchical clustering may be used to analyse existing monitoring networks; here we show how the same techniques, coupled with the output of a long-term simulation of an air-quality model, can provide an optimized network design (where we here define "optimized" as "having a common level of dissimilarity for potential station locations, for the dissimilarity metrics chosen"). Equivalently, these optimized networks maximize the dissimilarity, and hence minimize the potential redundancy, in the location of monitoring network stations.

Our first analysis using model output evaluates the extent to which the model is capable of creating similar clusters as the observations. Hourly model output for the one-year simulation of GEM-MACH was extracted from those model grid squares containing the station locations, and the resulting time series data were submitted to the same hierarchical clustering methodology as described above. Figure 7 shows the spatial distribution for the cluster analysis at the same levels of 1-R, 0.75, 0.65 and 0.55, as was shown using observation data (compare to Section 4, Figure 2). Each Airshed is plotted with a

different polygon, and colours indicate clusters. The corresponding dendrograms for these model results are shown in Supplement 1, Figure S14. Note that cluster colours/numbers differ between Figures 2 and 7; stations are falling within similar clusters in each Figure. For $SO_2$ dissimilarity level 1-R =0.75 (Figure 7a), the difference between the results for model and observations is not substantial; the clustering is identical aside from a single station both in WBEA and LICA, and AEP and PAS stations not forming separate clusters. The difference between observed and modelled $NO_2$ clustering

results is more notable as the level of dissimilarity decreases (Figure 7b,c): the model tends to create a larger number of clusters than the observations at intermediate levels of dissimilarity (comparing Figure 2b and Figure 7b: six clusters versus ten clusters; 2c and 7c: eleven clusters versus thirteen clusters). The model results also tend to cluster within the same Airshed to a greater degree compared to the observations results. The model dendrograms tend to have clusters forming at higher levels of dissimilarity for some stations such as Steeper (Figure S14 for Steeper is 1-R=0.8, while Figure S1 for

Steeper's node is 1-R=0.7). Some of these differences may be due to inaccuracies in the emissions data driving the model. For example, the major point source emissions data used in the simulations is based on regulatory reporting to the NPRI, wherein the regulatory requirement for reporting is an annual total. These annual totals must be temporally allocated using assumed temporal profiles for each source, and these month-of-year, day-of-week, and hour-of-day temporal profiles may not always match actual hourly emission levels at any given time. We show elsewhere (Akingunola *et al.*, 2017) that hourly

continuous emissions monitoring data used as model inputs may result in very different short-term concentration behaviour, with the corollary here that temporal allocation used here may influence the pattern of clusters. However, the model results at level of dissimilarity 0.65 tend to cluster more similarly with the observation results at level of dissimilarity at 0.55, indicating that the clustering analysis for the model results and observations show a similar spatial distribution, though the model shows overall higher correlation values than the observations.

The results for $SO_2$ (dendrograms for the cluster analysis in Supplement 1, Figure S12, compare to Figure S5) show the model results clustering similarly to the observations for PAMZ, ACCA and WCAS stations. Alternatively while WBEA stations in the model results (Figure S15, red station labels) are split into two clusters, while these stations are part of the same cluster in the observation-based analysis (Figure S5). At 1-R level 0.75, both model and observation cluster analysis results (Figure 8a, compare to Figure 4a) already show many clusters composed of one or few stations, with the model

showing slightly more clusters than the observations (21 clusters versus 25, respectively). As noted earlier, $SO_2$ in this region is emitted mainly by point-sources, and the use of annual emissions data with an assumed temporal allocation, along with the additional inherent difficulties in accurately predicting plume rise (Akingunola *et al.*, 2017), make the reproduction of the





time record of SO$_2$ by the model a challenge. Inaccuracies in both the emissions and the model meteorology may contribute to these differences.

We next show an example of how hierarchical clustering using gridded model output may be used to generate an optimized monitoring network. For this analysis, we focus on a specific sub-section of the model grid; namely a 72x72 block of model

grid-squares centred on the Athabasca Oil Sands. Figure 9 depicts the resulting mapped 1-R cluster analysis in this area, when each model grid-cell has been treated as a potential monitoring station location. Figure 9a,b shows the spatial distribution of the 1-R dissimilarity levels for NO$_2$ and for SO$_2$, respectively, and Figure 9c,d shows the spatial distribution of the clusters generated by dissimilarity levels of 0.65 for NO$_2$ and 0.8 for SO$_2$, respectively (these levels were chosen based on the analysis above, where the model was shown to provides reasonable results). All the panels in Figure 9 have the areas

where the oil and gas extraction sites and processing facilities are located as a visual aid; these areas are contoured in black.

The 1-R metric maps (Figure 9a,b) have the highest values where main emissions sources are located – these identify the main open-pit mine facilities of the oil sands, within which may be found both area and stack emissions sources. These regions of high variability are thus where the influence of the emissions and the local meteorology on the dispersion of the emissions is the strongest. In the NO$_2$ dissimilarity map "point" (stack), "line" (roads) and "area" sources (mines) can be

distinguished; for SO$_2$ the locations of the stacks for processing and flaring are identified. The spatial distribution of the *clusters* (each cluster is mapped with a different colour in Figure 9c,d shows the areas wherein a single measurement station, placed anywhere within a given coloured region, would represent that region to the given level of dissimilarity. Figure 9c thus shows that for NO$_2$, and for a 1-R dissimilarity level of 0.65, fourteen monitoring stations, each placed at any location within each of the fourteen coloured regions, would constitute an optimized network for NO$_2$. Similarly, Figure 9d shows

that seventeen stations would be required to monitor SO$_2$ with a common 1-R dissimilarity of 0.80, and the regions over which those stations could each be placed. The analysis thus identifies regions which are equivalent from the standpoint of the dissimilarity metric used.

We note that each of these coloured subregions in which a single station could be placed has a relatively large geographic extent, and, using this metric, do not describe the concentration gradient in the region. However, maps such as these could be

overlaid with other geographic information (e.g., road networks, the local power grid, etc.) to further optimize and decide on potential station locations. The similarity maps, combined with these other factors, could be used to aid in the design of air pollution monitoring networks.

The cluster distribution maps show that the areas for potential station location depend on the pollutant – the SO$_2$ map is influenced to a greater degree by the wind directions throughout the year than NO$_2$, likely due to the emissions sources for

the former pollutant being driven almost entirely by stack sources in this region. The wind-rose-like pattern around SO$_2$ sources likely stems from plume fumigation events at different times of the year, leading to a high correlation of SO$_2$ concentrations leading downwind from the sources. The NO$_2$ cluster distribution is patchier, reflecting both the impact of the stacks (which account for about 40% of the total NO emissions in the region) and the off-road mobile mine fleet (other "area" sources, which account for the bulk of the remainder of the NO$_x$ emissions). If potential multi-pollutant monitoring

station locations are desired, overlapping the optimized maps for each pollutant, for a given number of stations, would be a further way of aiding the monitoring network design process.

We also note that other metrics could be used in order to capture other aspects of concentration spatial and temporal variability, such as concentration gradients, in addition to temporal correlation – here we have demonstrated a "proof of concept", and other metrics will be analyzed in future work.



## 7    Potential factors impacting the analysis

Factors that can negatively impact the results of hierarchical clustering include data dispersion (large variance between cluster members), outliers and non-uniform cluster densities (clusters which are non-compact and non-isolated, thus not properly distinct from one another) (cf. Mangiameli *et al.*, 1996; Milligan, 1980). However, we find that the analysis itself may also be used to identify these conditions.  We have shown in the results in Section 4 and 5 that the analysis has indeed identified stations that are outliers relative to the rest of the dataset – these stations separate from the other stations as single-member clusters at high levels of dissimilarity. In other words, that is, the analysis identifies the records of those stations as being substantially different from all other station records, for the dissimilarity metric used.  This was particularly noticeable in the bimonthly data analyses. The methodology also identified cases of data dispersion, for example, the analysis of combined bimonthly passive and continuous monitors showed cases where monitors in close proximity or even collocated did not cluster together. The methodology thus seems capable of isolating outlier records and data dispersion, as well as recognizing cases of substantial differences between data collection methodologies. The latter was noted in the case of hourly ozone observations by Solazzo and Galmarini (2015).

The analysis of combined continuous and passive data has identified systematic differences between the two monitoring methodologies as a potential confounding factor on the station ranking of passive stations; the analysis identifies collocated stations with concentration differences and poorly matching concentration time variation, but cannot identify the causes for these differences. These issues should be the subject of follow-up work. Nevertheless, we note that both passive and continuous data may be subject to errors associated with the accuracy and precision of the sampling methodology.

We examined the potential errors associated with the reported detection limit of the monitoring methodology by using the GEM-MACH derived time series at station locations. Random noise was added to the original model time series results, with the maximum magnitude of the noise for each species taken from the detection limit range of each instrument (i.e. random noise in the range +/-0.5 ppbv was added to the $NO_2$ time series and +/-1 ppbv was added to the $SO_2$ time series). The $NO_2$ cluster results for hourly time series using 1-R as the dissimilarity metric (Figure S13, Supplement 1, compare to Figure 2), show no significant difference between the original and noise-added time series. However, this changed as time scales were removed from the original data sets by KZ filtering, especially once monthly and all shorter time scales were removed. Random noise was thus shown to be a potential confounding factor in 1-R hierarchical clustering analyses.  However, for the corresponding $NO_2$ Euclidean distance metric, both the hourly and monthly filtered data, with and without noise-added, resulted in identical clustering (not shown). The $SO_2$ results showed a larger variation between the clusters generated with the original time series and those containing additional random noise.  The difference in clustering was particularly noticeable for the 1-R dendrograms, for both hourly and time filtered data, and slightly less pronounced for Euclidean distances (not shown). The work described above suggests that much of the "signal" for primary emitted or quickly reacting secondary pollutants for correlation analysis resides in the shorter time scales (hourly to daily); the greater influence of random noise on the results of the time-filtered data implies that the latter are dominated by close-to-background concentrations, which are in turn similar in magnitude to the noise levels added here, and hence a greater influence is seen on clustering of the time-filtered data. For species such as $SO_2$, which are dominated by short-duration high concentration plumes, this effect may extend to the shorter timescales as well.

As mentioned in Sollazzo and Gamarini (2015), the manner in which the data is reported may significantly impact the analysis. Besides the detection limit of the instrument, Airsheds report the passive observations with a reporting limit of 0.1 ppbv, hence we also tested the accuracy of the instrument or the number of significant figures being reported, again using model time series at station locations as a surrogate for observation data. The model results were filtered for three or zero significant figures below the decimal, and the resulting analyses were compared. As for the random error test, we found that for both $NO_2$ and $SO_2$ the dendrogram patterns changed, indicating that the use of fewer significant digits in data reporting will result in enough loss of information to change the interpretation of the data.





In the analysis described in Section 4, it was noted that as successively larger time scales are filtered from the data used for clustering, the magnitudes of the clustering metrics show an increasingly higher degree of similarity, with monitors clustering both within and across Airsheds. However, the *filtering* of time series to remove successively larger time scales is not equivalent to *averaging*, in which shorter time scale information may be retained in the average. To specifically examine

the effect of time averaging during data collection on clustering results, the clusters for the hourly data were compared to those from daily, weekly and monthly averages (Figure 10). With the original hourly data, specific Airsheds were identified as unique clusters (as expected, for 1-R hierarchical clustering; stations located close to Airshed-specific sources were identified as being more similar). However, with increasing averaging times, this Airshed-specific clustering was gradually lost. Most of the information driving the ability of 1-R clustering to link local sources was thus shown to reside in the shorter

time scales. Nevertheless, this information was lost as increasing *averaging* periods were applied (Figure 10). A fundamental result of this analysis is that measurements that consist of long-term averages may lose the ability to identify the influence of local sources on the basis of time variation, i.e., they will correlate at an equal level with both adjacent monitoring stations and those that are located in distant regions. However, this information is retained in hourly records, and the latter may be used to identify unique source regions on the basis of correlation.

**8    Conclusions**

A methodology for cross-comparing air quality monitoring networks was proposed here, expanding on the work of Solazzo and Galmarini (2015) by including the Euclidean distance as well 1-R as dissimilarity metrics for hierarchical clustering, and by making use of chemical reaction-transport model output as a surrogate for observation station data. We adopted the KZ filter in its original low-pass configuration, in order to improve the ability of the methodology to distinguish the impact

of different time scales of variation on clustering. The Euclidean distance metric allowed cross-comparison of the stations in terms of the magnitude of the concentrations, whereas 1-R evaluated their temporal variation similarity. Both metrics can be used together or separately to evaluate the similarity of the stations and their potential redundancy. The relative level of potential redundancy for existing observation stations was ranked based on each dissimilarity metric, and we recommend evaluating monitoring station redundancy using both metrics where possible. Stations which form clusters at low values of

both 1-R and Euclidean distance are the most redundant, while those with high values of either or both of these metrics are the least redundant. Absolute thresholds for redundancy cannot be generated since the relative rankings depend on the available observation data (number of stations and chemical species observed). In addition, other considerations such as spatial proximity to sensitive receptors, the regulatory purpose of the station(s), and logistics (e.g. accessibility or power supply), may outweigh the recommendations based on similarity alone.

We have shown, through several analyses, that much of the observation signal which may be used to identify common sources of both primary pollutants and secondary products of fast reactions resides in shorter time scales (hourly to daily). When hourly data are available, the methodology is able to identify groups of stations that are influenced by common emissions sources (e.g., stations that are influenced by oil sands emissions as opposed to stations located elsewhere), as well as identify outliers or stations records that are markedly different from all others in a given dataset. The former property is

useful for identifying the influence range of specific emission sources. The latter property shows that the methodology is a useful tool for identifying station instrumentation that may be located such that they are subject to unique conditions (e.g. very nearby sources, anomalous long-term variation, etc.), or which have anomalous readings. However, for data consisting of longer-term averages, or observations in which the shorter time scales have been removed by filtering, at least some of the information which identifies the influence of common emissions sources is lost. Nonetheless, the methodology, when

applied to time-filtered data, is able to single out stations mainly influenced by seasonality.





Clustering was shown to depend on the chemical species analyzed, suggesting that optimization of networks using this methodology should be carried out on a "by species" basis rather than a "by station" basis. The two species examined here originate in different types of emissions sources in the region under study, and consequently have different dissimilarity rankings for the corresponding stations.

We have corroborated the work of Solazzo and Galmarini (2015) for ozone in that the methodology is capable of identifying monitoring stations making use of different monitoring methodologies (via our 5 year analysis of passive and continuous $SO_2$ and $NO_2$ observations on a common bimonthly averaging interval). Passive and continuous monitors in the same airsheds did not always fall within common clusters (with several examples in which collocated monitors from the two technologies did not correlate). Some of these issues may be result of averaging time, though data round-off and accuracy

(random noise) were also shown to have a negative influence on the clustering results.

We have expanded the use of hierarchical clustering for air pollution to include its use with air-quality model output. This presents a new avenue for monitoring network optimization and design in that each high resolution air-quality model grid square can be treated as a potential monitoring station location. Comparisons of the results of the clustering of model and observed time series at monitoring station locations showed clusters generated from model output tended to be more similar

within Airsheds than was the case for clusters generated from observations. However, the results are quite comparable, albeit at higher correlation levels for the model than the observations, and the match to observations depends on the chemical species. Tests in which gridded model output were treated as potential station locations resulted in the first dissimilarity analysis based maps of optimized air pollution monitoring networks. These showed that the methodology is capable of generating sub-regions within which a single station will represent that entire sub-region, to a given level of a dissimilarity

metric. Maps of this nature may be combined with other georeferenced data (e.g., road networks, power availability) to assist in monitoring network design.

While hierarchical clustering's pitfalls include data dispersion and outliers, we show here that the methodology is also able to identify differences in sampling methodologies and anomalous stations records. The analysis was shown to be particularly sensitive for monitors sampling air contaminants such as $SO_2$, in areas of low background concentrations and sudden

concentration peaks. For $SO_2$, this is a result of the variation inherent in the type of sources that dominate $SO_2$ emissions in our study region, i.e., large stack plumes. We also note that comparing observation-based cluster analysis with those of air-quality model output at station locations might help identify possible deficiencies in the emission data used to drive air quality models. Given that short-term variation has been shown here to have a key impact on identifying common sources, the use of annual totals and assumed temporal profiles as the basis for emission inventory reporting should be avoided, and

more time specific records, should be used where possible.

## 9   Author Contribution

JS: Study concept and design, applying the methodology, analysis of the cluster analysis results, and writing of manuscript and modifications of same; P.A.M: Study concept and design, analysis of the cluster analysis results, and writing of manuscript and modifications of same; Y.A.: providing QC/QA AEP air quality monitoring data and data description; A.A.:

GEM-MACH simulations. In addition, the first author would like to thank all co-authors for extensive comments on different versions of the manuscript.

## 10   Competing interests

The authors declare that they have no conflict of interest.



## 11    Acknowledgements

This project was jointly supported by the Climate Change and Air Quality Program of Environment and Climate Change Canada, Alberta Environment and Parks, and the Joint Oil Sands Monitoring program.   The figures in this work were created using a combination of Environment Canada and Climate Change software 
and the R open-source programming language (R Core Team, 2017).

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





**Tables**

Table 1 Hourly NO$_2$ Similarity Ranking for the 1-R and Euclidean Distance (EuD) metrics. Note that stations at the bottom of the two columns are the most similar (hence one measure of their level of redundancy) with respect to each metric of dissimilarity. Here we show only the first 10 and last 10 items of the ranking, the full ranking can be consulted in Table S5 in Supplement 1.

| 1-R | Name | ID | Aished | EuD | Name | ID | Aished |
|-----|------|-----|--------|-----|------|-----|--------|
| 0.72 | Maskwa | 1248 | LICA | 1009 | Shell Muskeg River | 1244 | WBEA |
| 0.61 | Anzac | 1225 | WBEA | 950 | Millennium Mine | 1075 | WBEA |
| 0.60 | ST.LINA | 1250 | LICA | 950 | Fort McMurray-Athabasca Valley | 1064 | WBEA |
| 0.56 | Steeper | 1055 | WCAS | 923 | Grande Prairie (Henry Pirker) | 1165 | PAZA |
| 0.56 | Caroline | 1092 | PAMZ | 839 | Calgary Northwest | 1039 | CRAZ |
| 0.55 | Lethbridge | 1049 | AEP | 839 | Calgary Central 2 | 1221 | CRAZ |
| 0.55 | Crescent Heights | 1172 | PAS | 807 | Redwater Industrial | 1156 | FAP |
| 0.54 | Wagner2 | 1241 | WCAS | 769 | Red Deer-Riverside | 1142 | PAMZ |
| 0.54 | Genesee | 1057 | WCAS | 735 | Edson | 1062 | WCAS |
| 0.51 | Shell Muskeg River | 1244 | WBEA | 722 | Meadows | 1058 | WCAS |
| … | … | … | … | … | … | … | … |
| 0.18 | Range Road 220 | 1161 | FAP | 400 | Fort Saskatchewan | 2001 | FAP |
| 0.16 | Lamont County | 1162 | FAP | 387 | Anzac | 1225 | WBEA |
| 0.16 | Elk Island | 1157 | FAP | 350 | Violet Grove | 1052 | WCAS |
| 0.16 | Fort McKay South | 1076 | WBEA | 350 | Tomahawk | 1053 | WCAS |
| 0.16 | Fort McKay-Bertha Ganter | 1032 | WBEA | 348 | Power | 1059 | WCAS |
| 0.15 | Edmonton Central | 1028 | ACCA | 301 | Caroline | 1092 | PAMZ |
| 0.14 | Woodcroft | 2002 | ACCA | 280 | Steeper | 1055 | WCAS |
| 0.14 | Edmonton South | 1036 | ACCA | 280 | ST.LINA | 1250 | LICA |
| 0.11 | Ross Creek | 1159 | FAP | 263 | Lamont County | 1162 | FAP |
| 0.11 | Fort Saskatchewan | 2001 | FAP | 263 | Elk Island | 1157 | FAP |

Table 2 Hourly SO$_2$ Similarity Ranking. Note that stations at the bottom of the two columns are the most similar (hence one measure of their level of redundancy) with respect to each metric of dissimilarity. Here only the first and last 10 items of the ranking, the full ranking can be consulted in Table S6 in Supplement 1.

| 1-R | Name | ID | Aished | EuD | Name | ID | Aished |
|-----|------|-----|--------|-----|------|-----|--------|
| 1.01 | Redwater Industrial | 1156 | FAP | 1594 | Redwater Industrial | 1156 | FAP |
| 0.95 | Caroline | 1092 | PAMZ | 709 | Mannix | 1069 | WBEA |
| 0.88 | Valleyview | 1170 | PAZA | 532 | Mildred Lake | 1066 | WBEA |
| 0.88 | Smoky Heights | 1167 | PAZA | 470 | Millennium Mine | 1075 | WBEA |
| 0.85 | Maskwa | 1248 | LICA | 412 | Shell Muskeg River | 1244 | WBEA |
| 0.85 | Mannix | 1069 | WBEA | 372 | Lower Camp | 1074 | WBEA |
| 0.83 | Red Deer-Riverside | 1142 | PAMZ | 269 | CNRL Horizon | 1226 | WBEA |
| 0.81 | Steeper | 1055 | WCAS | 231 | Wagner2 | 1241 | WCAS |
| 0.81 | Power | 1059 | WCAS | 231 | Genesee | 1057 | WCAS |
| 0.81 | Meadows | 1058 | WCAS | 220 | Edmonton East | 1029 | ACCA |



| | | | | | | | |
|---|---|---|---|---|---|---|---|
| 1.01 | Redwater Industrial | 1156 | FAP | 215 | Maskwa | 1248 | LICA |
| … | … | … | … | … | … | … | … |
| 0.48 | Wagner2 | 1241 | WCAS | 102 | Caroline | 1092 | PAMZ |
| 0.48 | Genesee | 1057 | WCAS | 91 | Smoky Heights | 1167 | PAZA |
| 0.45 | Range Road 220 | 1161 | FAP | 79 | Carrot Creek | 1054 | WCAS |
| 0.45 | Fort Saskatchewan | 2001 | FAP | 70 | Lethbridge | 1049 | CRAZ |
| 0.39 | Lamont County | 1162 | FAP | 58 | Beaverlodge | 1168 | PAZA |
| 0.39 | Bruderheim | 2000 | FAP | 55 | Grande Prairie (Henry Pirker) | 1165 | PAZA |
| 0.35 | Fort McMurray-Patricia McInnes | 1070 | WBEA | 50 | Crescent Heights | 1172 | PAS |
| 0.35 | Fort McMurray-Athabasca Valley | 1064 | WBEA | 42 | Evergreen Park | 1166 | PAZA |
| 0.19 | Fort McKay South | 1076 | WBEA | 24 | Steeper | 1055 | WCAS |
| 0.19 | Fort McKay-Bertha Ganter | 1032 | WBEA | 24 | Red Deer-Riverside | 1142 | PAMZ |



**Figures**

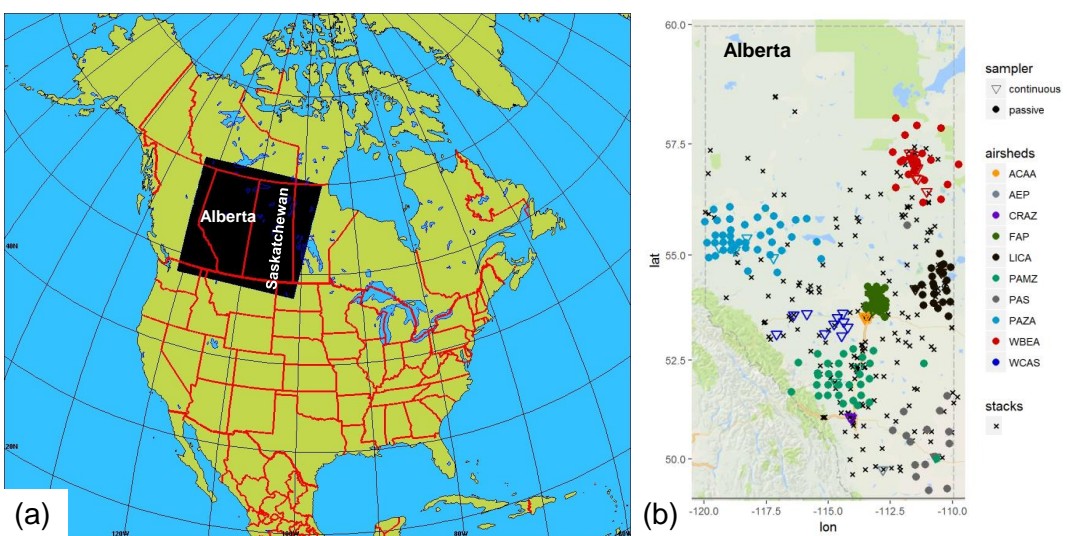

Figure 1: Study area: a) model domain covering the provinces of Alberta and Saskatchewan, and b) NO$_2$ and SO$_2$ continuous and passive monitors located at the different air quality monitoring networks (Airsheds) and main NO$_2$ and SO$_2$ stacks in the Province of Alberta. Stations are colour-coded according to Airsheds and plotted with different polygons (circle for passive, inverted triangle for continuous): West Central Airshed Society (WCAS), Wood Buffalo Environmental Association (WBEA), Fort Air Partnership (FAP), Alberta Capital Airshed Alliance (ACAA), Calgary Regional Airshed Zone (CRAZ), Peace Airshed Zone Association (PAZA), Palliser Airshed Society  (PAS), Parkland Airshed Management Zone (PAMZ) and Lakeland Industrial Community Association (LICA).

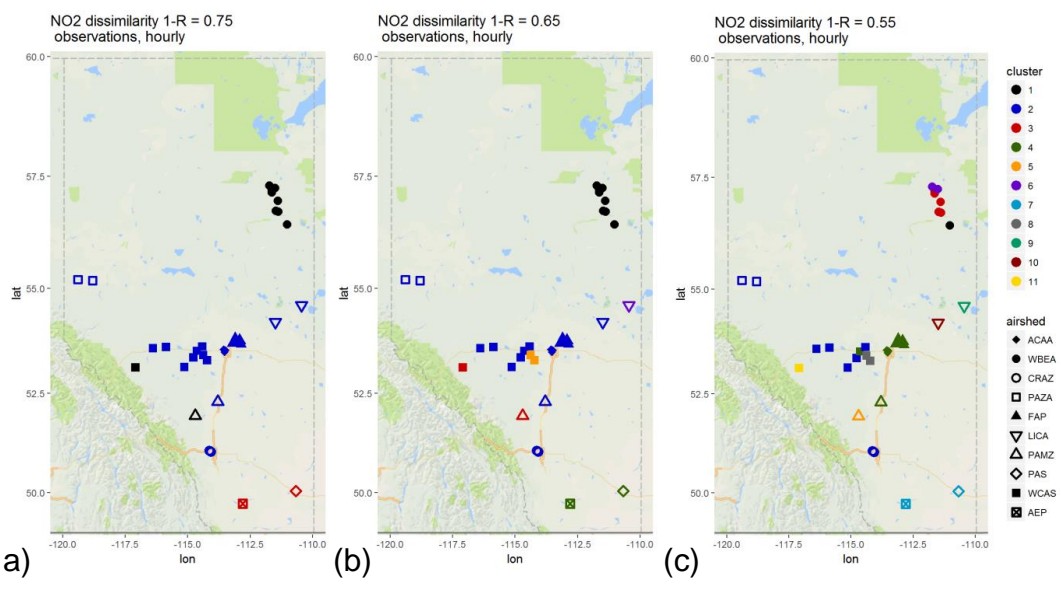

Figure 2: Associativity analysis for observed NO$_2$ hourly time series using 1-R as the metric to compute the dissimilarity matrix, assuming a dissimilarity level of a) 0.75, b) 0.65 and c) 0.55. Stations are colour-coded by cluster, and Airsheds are plotted with different polygons. The acronyms for the Airsheds are as in Figure 1.




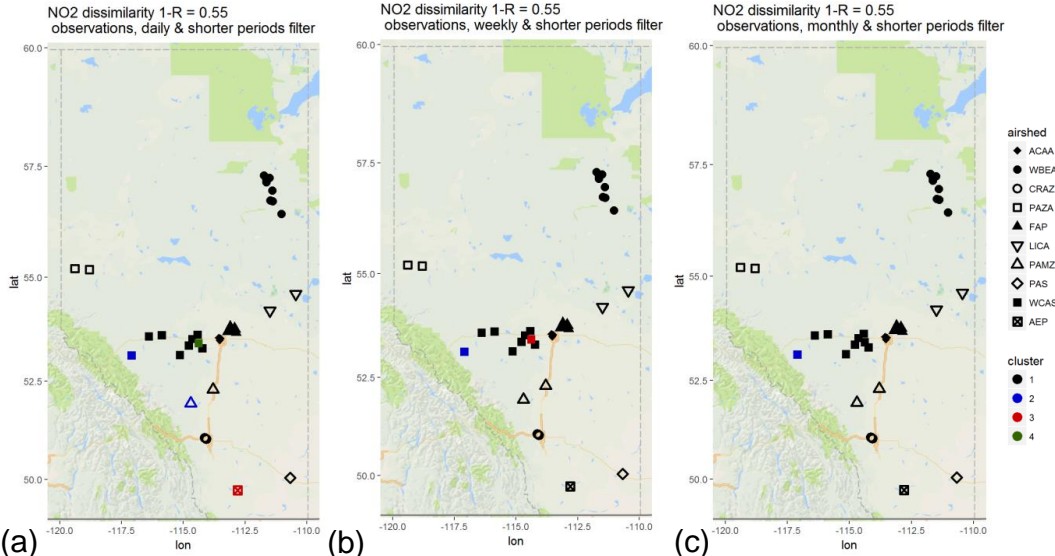

**Figure 3: Associativity analysis for observed NO$_2$ filtered time series using 1-R as the metric to compute the dissimilarity matrix, assuming a dissimilarity level of 0.55: a) daily, b) weekely and c) monthly and short time periods. Stations are colour-coded according to cluster formation, and Airsheds are plotted with different polygons. The acronyms for the Airsheds are as in Figure 1.**

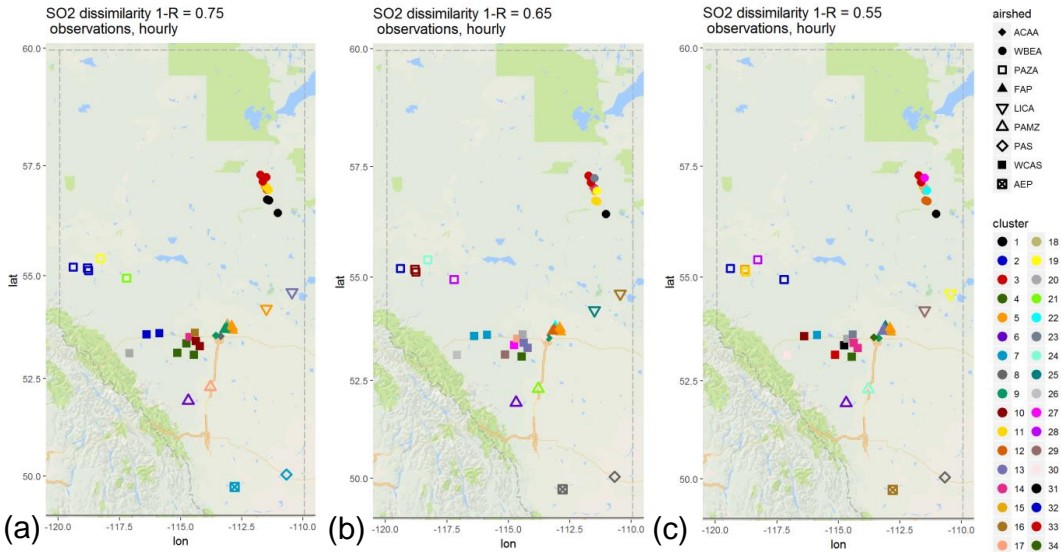

**Figure 4: Associativity analysis for observed SO$_2$ hourly time series using 1-R as the metric to compute the dissimilarity matrix, assuming a dissimilarity level of a) 0.75, b) 0.65 and c) 0.55. Stations are colour-coded by cluster, and Airsheds are plotted with different polygons. The acronyms for the Airsheds are as in Figure 1.**





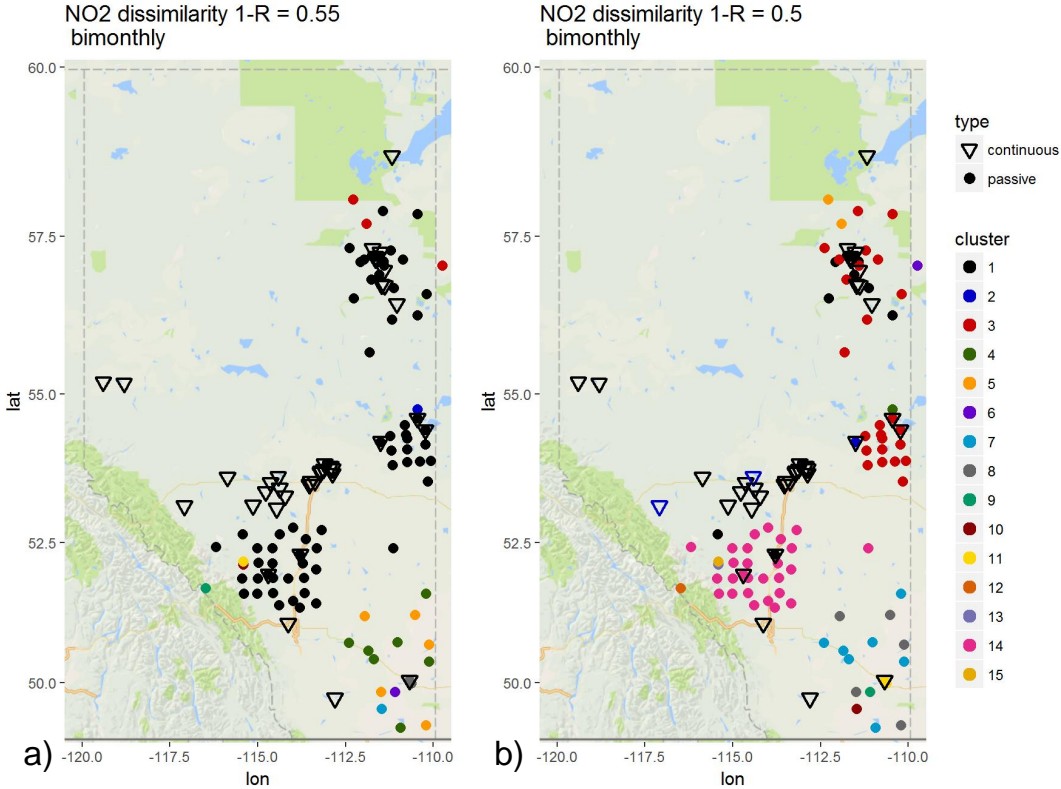

**Figure 5:** Associativity analysis for passive and continuous bimonthly $NO_2$ averages for 1-R = 0.55 (R=0.3) Stations are colour-coded according to cluster formation, with continuous stations are marked as inverted triangles and passive stations as circles. The acronyms for the Airsheds are as in Figure 1.




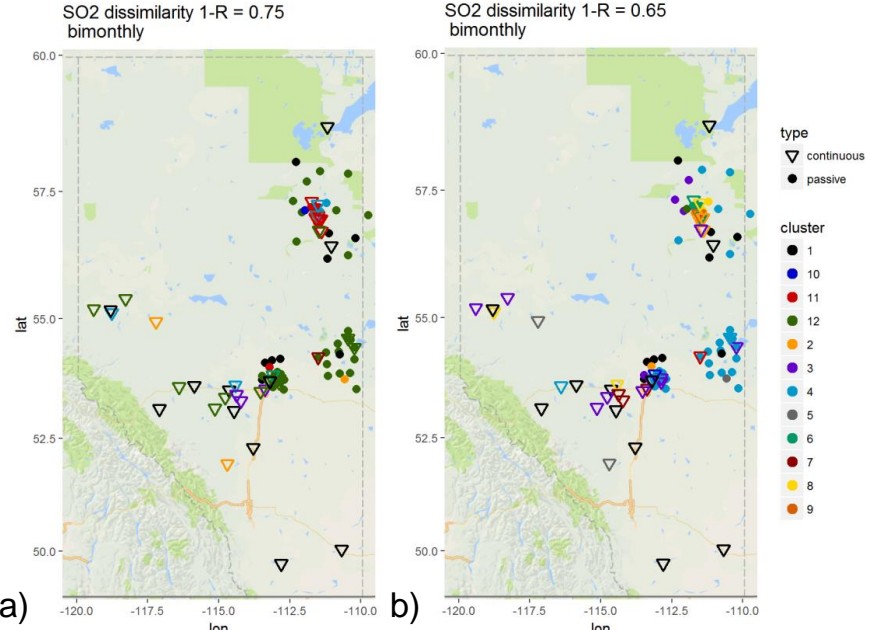

**Figure 6: Associativity analysis for passive and continuous bimonthly SO₂ averages for 1-R = 0.7 (R=0.3) Stations are colour-coded according to cluster formation, with continuous stations are marked as triangles and passives as circles. The acronyms for the Airsheds are as in Figure 1.**

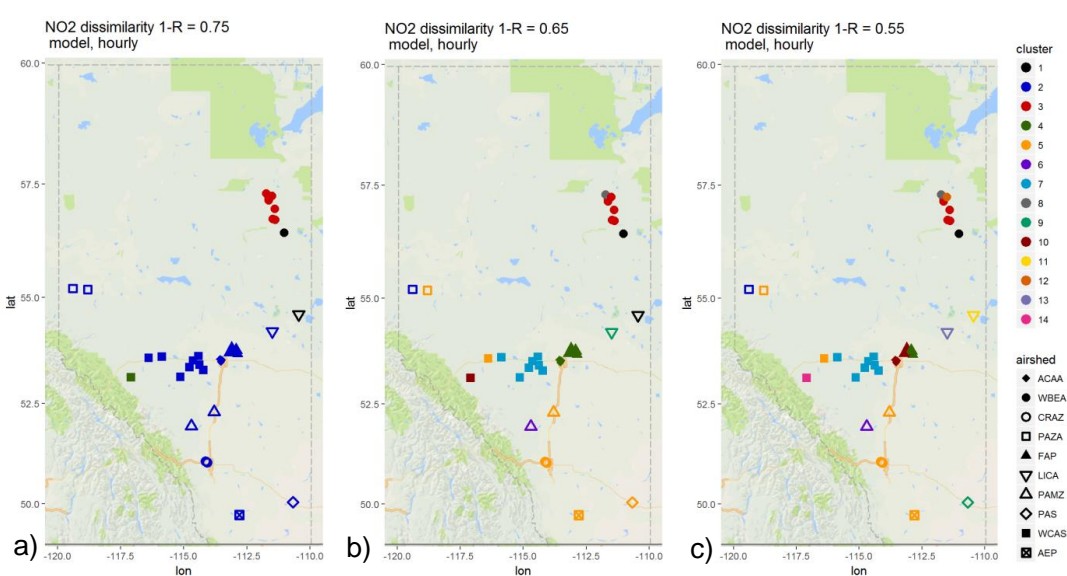

**Figure 7: Associativity analysis for modelled NO₂ hourly time series using 1-R as the metric to compute the dissimilarity matrix, assuming a dissimilarity level of a) 0.75, b) 0.65 and c) 0.55. Stations are colour-coded according to cluster formation, and Airsheds are plotted with different polygons. The acronyms for the Airsheds are as in Figure 1.**





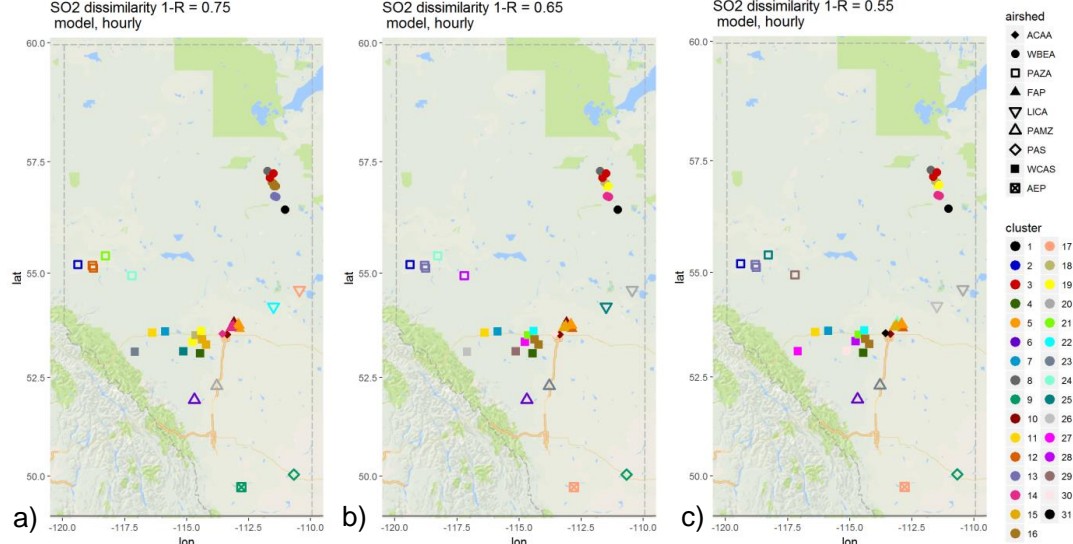

**Figure 8: Associativity analysis for modelled SO₂ hourly time series using 1-R as the metric to compute the dissimilarity matrix, assuming a dissimilarity level of a) 0.75, b) 0.65 and c) 0.55. Stations are colour-coded according to cluster formation, and Airsheds are plotted with different polygons. The acronyms for the Airsheds are as in Figure 1.**



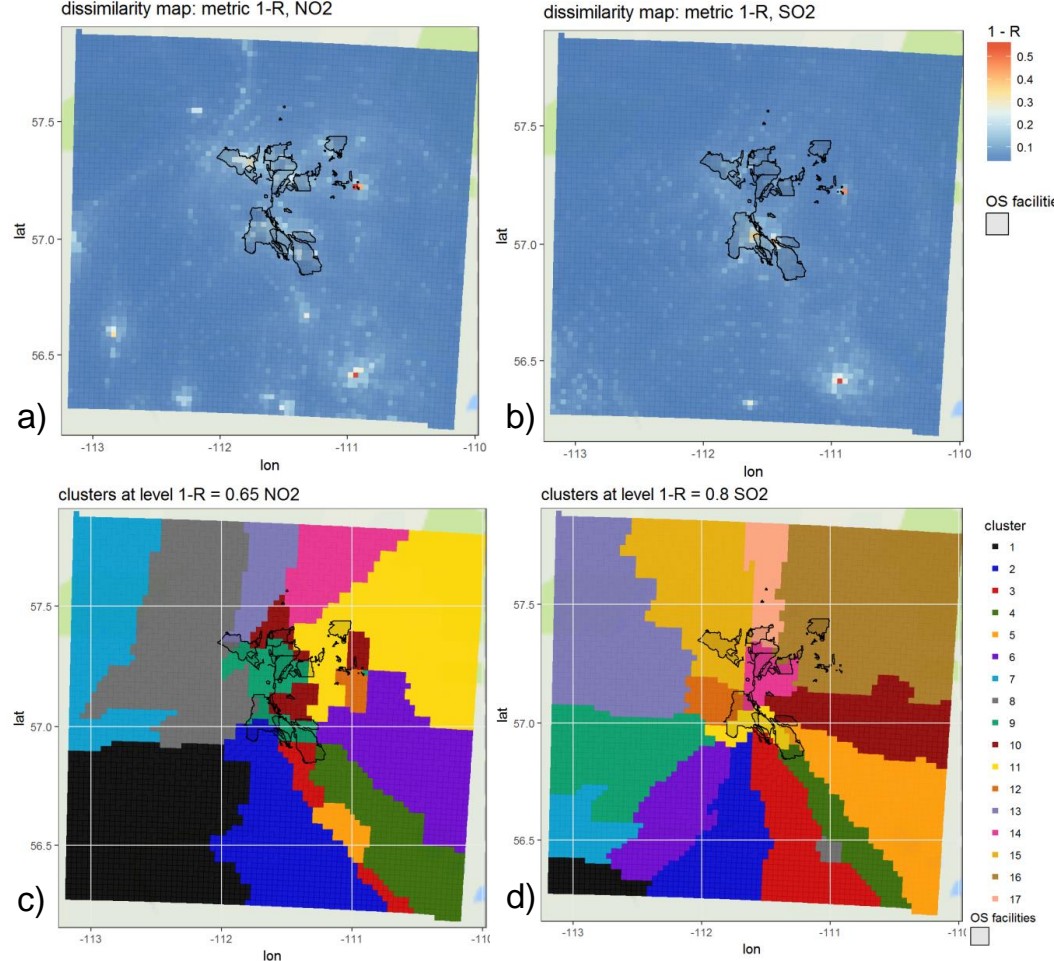

**Figure 9: Dissimilarity maps based on 1-R metric for a) NO₂ and b) SO₂ modelled hourly output at each GEM-MACH grid-cell. Associativity analysis maps for modelled NO₂ and SO₂ based on these gridded output time series, appear in c) and d), respectively. The latter maps were generated using a (1-R) dissimilarity level of b) 0.65, and d) 0.8. All maps show the areas enclosing the property boundaries of the main mining facilities operating in the Athabasca oil sands region (black contours enclosing transparent light grey shading).**



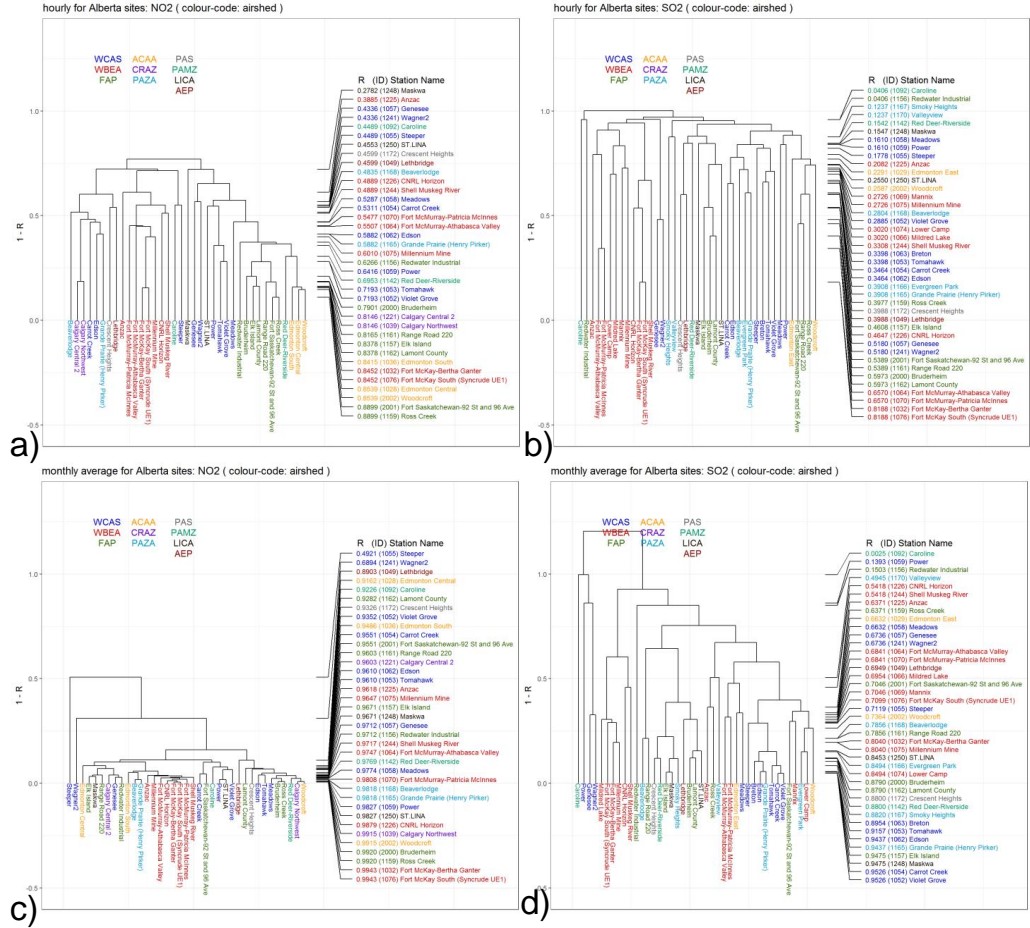

Figure 10: Dendrogram analysis for NO$_2$ and SO$_2$ hourly (a) and b), respectively) and monthly or shorter time scales time series (c) and d), respectively) using 1-R as the metric to compute the dissimilarity matrix, for the Airsheds decribed in Figure 1. The dendrongram is colour-coded according to Airshed. Right side:stations ranked from low to high correlation level.