# Peer review of "The Use of Hierarchical Clustering for the Design of Optimized Monitoring Networks"

_Atmospheric Chemistry and Physics, 2017_

## Referee Comment (RC1) · Anonymous Referee #1 · 6 Feb 2018

Review of acp-2017-1126. Associativity Analysis of SO2 and NO2 for Alberta Monitoring Data Using KZ Filtering and Hierarchical Clustering

The papers is well written, scientifically sound, and clearly reflects the large amount of work behind the analysis as well as the authors' knowledge of the relevant scientific literature. The presentation is clear and overall reads really well. In general, the application of data mining techniques to air quality monitoring and modeling is still at an 'embryonic stage' and deserves encouragement.

I advice the editor to accept the manuscript for publication. I only have some minor suggestions on my notes for the authors to consider including in the final version.

[Figure]

General and specific comments:

1) Given the scientific significance and the potentiality of this work, I believe it deserves more visibility. I think the authors are underselling their work. For instance, the title seems to suggest a study with highly technical details which can discourage non-expert readers, whilst could be more general to attract more audience. Consider avoiding the use of KZ in the title, it is just a moving average filter.

2) It's not clear to me the average behind figure 9. It shows the correlation map of each grid cell with any other cell? Does it imply an average over R? or it is the time or spatial series correlation being investigated? Please clarify in the text

3) it is not clear how redundancy is defined: Overlapping variance, coefficient of determination above a certain threshold, ...;

4) based on this study, can the authors comment on the minimum exposure period (length of time series) for the clustering analysis to be reliable

5) page 4 ,line 17. Consider Vardoulakis et al. 2011. Atmospheric Environment 45 (2011) 5069-5078

6) page 8, line3. Not only 'dissimilarity metrics' but also agglomeration method and definition of correlation coefficient are quite sensitive parameters

7) Can the Euclidean distance be used to spot systematic detection error?

8) page 14 line 16. Can the authors comment on the spatial continuity of the solution? Is it a requirement or the area can contain holes and/or be even detached?

9) Page 15, line37. Solazzo and Galmarini misspelled.

10) Page 15, line37. A source of dissimilarity was found to be the reporting time not harmonised across European countries. Data reporting at the beginning or at the end of the hour can make a significant difference

I invite the authors to comment on the following:

I think we are still far away from using clustering for operational use. Clustering is known to provide some qualitative insight, but it is quantitatively weak as it depends on many parameters. Indeed, a fundamental challenge of clustering is the high sensitivity to the options controlling the underlying algorithms, such as the agglomerative method, the distance metric, the number of clusters, and the cut-off distance are aspects that need to be determined case by case. In particular, the cut-off (the threshold similarity above which clusters are to be considered disjointed) determines the dimension of the sub-space of non-redundant information and is decided by visual inspection of the dendrogram. Supervised clustering (e.g. k-Means) initiated with the results of unsupervised clustering might be more robust.

The application of associativity analysis for detecting potential redundancy in the context of regulatory air quality monitoring might have some pitfalls (most of which are anyway mentioned by the authors in the text, but I think deserve more words). For example, the potential duplicate of information obeys some policy precautionary principle and might reveal useful in some instances (double checking, reduce missing records, cross validation, etc). Further, redundancy should be determined with some long-term climatology and should also serve future decision making in the sense that what might be redundant based on the past ten years of data might not be in the next ten years. In this sense the adoption of models for future scenarios might help.

I think that, more than the estimation of redundancy, the main strengths of the methodology are the potential for classification and the estimation of the area of representativeness (AoR). Indeed I would have framed the whole work in the context of classification. For example, can the methodology assist in the classification of monitoring station based on area-type or site-type? Do the authors expect the diurnal signal to be the associated over long distance? In siting a new station, its area-type can be defined by looking at how the signal of existing stations compares with the signal of the new station? I would invite the authors to add some further considerations about the

potentiality of the methodology devised, also in light that some reflections are already part of the paper, for example the clustering of long term signals.

Concerning the AoR, the authors (or at least some of them) have already experience with the topic, and I have been surprised that it was not expanded in the text, especially since model results are available. The maps in figure 9 indeed show some AoR! The authors mentioned it at the beginning of page 3 but then drop it. For example, some discussion about AoR would fit nicely in section 4.1. Again, in light of better exploiting the large amount of work done, I would invite the authors to consider adding some further words about the potentiality of the analysis for determining the AoR.

---

## Referee Comment (RC2) · Anonymous Referee #2 · 19 Feb 2018

I feel like congratulating the authors for the nice piece of work that though being an application of a technique developed by others, it also includes an expansion of the latter with the use of alternative metrics and model calculations.

The application case is very well suited for testing this technique given the density of samplers in the region analyses, the two kinds of samplers, the large number of airsheds and operating networks.

In my view, the work fulfils the journal standards of scientific originality and relevant and is also well written. I see no objection to its publication in its current status.

---

## Referee Comment (RC3) · Anonymous Referee #3 · 20 Feb 2018

The paper nicely describes the application of KZ filtering and hierarchical clustering of SO2 and NO2 data, both from observed and modelled time series in Alberta region. The paper is very well written and structured and easy to follow, although i recommend the authors to follow the classical structuring under introduction, materials and methods, results , etc... I think the paper is suitable for publication in ACP with few minor comment i raise below.

1) The abstract is too long and can be shortened only giving the key results and a recommendation to follow.

2) Can the authors explain why they consider only SO2 and NO2?

[Figure]

3) In the introduction, between lines 25-39, the authors only list the available literature but do not make a synthesis of these results and link it to their motivation of doing this study. What was missing in these studies?

4) Is it not possible to higher in resolution in the modelling part as 2.5 km resolution might be coarse for the purpose of the study? I think this deserves a discussion.

5) Figure title of S6, S7 and S8 are wrong, please correct them to SO2.
* * *

---

## Author Comment (AC1) · 16 Mar 2018

Author's reply to peer-review comments on"Associativity Analysis of SO2 and NO2 for Alberta Monitoring Data Using KZ Filtering and Hierarchical Clustering" by Joana Soares et al. submitted to ACP

Dear Anonymous Referee #2, We are grateful for your efforts and for the very positive evaluation of our manuscript.

---

## Author Comment (AC2) · 16 Mar 2018

**Author's reply to peer-review comments on**
"Associativity Analysis of SO2 and NO2 for Alberta Monitoring Data Using KZ Filtering and Hierarchical Clustering" by Joana Soares et al. submitted to ACP

Dear Anonymous **Referee #3**,

We are grateful for your efforts and the overall positive evaluation of our manuscript. The constructive comments have helped us to further improve our paper. Below we give our detailed responses to your comments and describe the revisions prepared for the manuscript. The Referee comments are cited in italics and our responses in regular type while revisions prepared to the manuscript are marked in red.

General and specific comments:
*1) The abstract is too long and can be shortened only giving the key results and a recommendation to follow.*

- The Authors have shorted the abstract: "Associativity analysis is a powerful tool to deal with large-scale datasets by clustering the data on the basis of (dis)similarity, and can be used to assess the efficacy and design of air-quality monitoring networks. We describe here our use of Kolmogorov-Zurbenko filtering and hierarchical clustering of $NO_2$ and $SO_2$ passive and continuous monitoring data, to analyse and optimize air quality networks for these species in the province of Alberta, Canada. The methodology applied in this study assesses dissimilarity between monitoring station time series based on two metrics: 1-R, R being the Pearson correlation coefficient, and the Euclidean distance; we find that both should be used in evaluating monitoring site similarity. We have combined the analytic power of hierarchical clustering with the spatial information provided by deterministic air quality model results, using the gridded time series of model output as potential station locations, as a proxy for assessing monitoring network design and for network optimization. ~~We find that both metrics should be used to evaluate the similarity between monitoring time series, since this allows a cross-comparison in terms of temporal variation and magnitude of concentrations to assess station potential redundancy. Here, the relative level of potential redundancy of an existing monitoring location was ranked according to each dissimilarity metric, with sites forming clusters at low values of both 1-R and Euclidean distance being the most redundant.~~ We demonstrate clustering results depend on the air contaminant analyzed, reflecting the difference in the respective emission sources of $SO_2$ and $NO_2$ in the region under study. Our work shows that much of the signal identifying the sources of $NO_2$ and $SO_2$ emissions resides in shorter time scales (hourly to daily) due to short-term variation of concentrations, and that longer term averages in data collection may lose the information needed to identify local sources. However, the methodology  identifies stations mainly influenced by seasonality, if larger time scales (weekly to monthly) are considered.  We have performed the first dissimilarity analysis based on gridded air-quality model

output, and have shown that the methodology is capable of generating maps of sub-regions within which a single station will represent the entire sub-region, to a given level of dissimilarity.  We have also shown that our  approach is capable of identifying different sampling methodologies, as well as identifying outliers (stations' time series which are markedly different from all others in a given dataset)."

*2) Can the authors explain why they consider only SO2 and NO2?*

- This manuscript focused only on $NO_2$ and $SO_2$ because only these two species had both passive and continuous monitoring data available, as mentioned in P3, L34-35 "We analyse data from both passive and continuous instruments measuring $NO_2$ and $SO_2$ ambient concentrations, the two species that include observations from both measurement methodologies." We have examined other continuous data using the methodology, and intend to discuss these other air contaminants in future work.  We revised the text to make this clearer in the manuscript, viz:

  P3,L34-35 "In this study we included observations  from both passive and continuous instruments measuring $NO_2$ and $SO_2$ ambient concentrations, since these are the only two species in the available data that include observations from both of these measurement methodologies."

*3) In the introduction, between lines 25-39, the authors only list the available literature but do not make a synthesis of these results and link it to their motivation of doing this study. What was missing in these studies?*

- The authors wanted to describe the scientific work using cluster analysis of observational data that apply the same metrics used in this study. We are not implying that is missing something in the referenced work, we wanted to illustrate how cluster analysis techniques have been used for different species and locations.  The text was revised to accommodate this comment.

  P2, L33: "oxidant ($O_x$), non-methane hydrocarbons (NMHC), and PM. In this past work, cluster analysis is usually applied to a small number of stations (5 to 70) in different locations around the globe. Solazzo and Galmarini (2015) applied cluster analysis data pre-filtered by iterative moving averages (Kolmogorov-Zurbenko (KZ) filtering, Zurbenko, 1986). Their work showed that cluster analysis can potentially accommodate different sampling technologies, and could be applied for large areas without the need of prior knowledge of the study area.  Here, Solazzo and Galmarini (2015) applied cluster analysis data pre-filtered "

  P3, L4: "(2015) and references therein, and further expands that methodology to focus on monitoring network optimization. We use the methodology for the first time for observation datasets collected in Alberta, analysing the data using two different similarity metrics, and rank existing observation stations based on relative station redundancy.  We then extend the methodology to a new application of gridded air-quality model data – showing that time series from a deterministic air quality model (Global Environmental Multiscale – Modelling Air-quality and Chemistry; GEM-MACH) may be used as a surrogate for observations in air-quality clustering analysis. . Dissimilarity may thus be used to rank stations in terms of

potential redundancy, where stations having the lowest levels of dissimilarity may be considered sufficiently similar to be considered potentially redundant.

 The combined use of deterministic model output and clustering analysis is shown to be a potentially powerful tool for network design, and/or optimization of existent air quality networks.

*4) Is it not possible to higher in resolution in the modelling part as 2.5 km resolution might be coarse for the purpose of the study? I think this deserves a discussion.*

- The potential use of even higher resolution (1km) was examined in separate work. The results were inconclusive in that higher resolution does not guarantee a more accurate air-quality forecast. For example, if the predicted synoptic or mesoscale meteorology is inaccurate due to poor spatial representation of a region in the meteorological monitoring network, then the benefits of higher resolution in air-quality simulations (resolving the sources to a higher degree) may be overwhelmed by the issues associated with highly resolved plume locations being inaccurately predicted. There are also practical computational considerations – to carry out the same domain simulations as carried out here would have required a 6.25x increase in processing time and memory.

*5) Figure title of S6, S7 and S8 are wrong, please correct them to SO2.*

- The authors noted that the dendograms are actually for $NO_2$ and not for $SO_2$, as it should be. The figures were revised.

---

## Author Comment (AC3) · 16 Mar 2018

**Author's reply to peer-review comments on**

"Associativity Analysis of SO2 and NO2 for Alberta Monitoring Data Using KZ Filtering and Hierarchical Clustering" by Joana Soares et al. submitted to ACP

Dear Anonymous **Referee #1**,

We are grateful for your efforts and the overall positive evaluation of our manuscript. The constructive comments have helped us to further improve the paper. Below we give our detailed responses to your comments and describe the revisions prepared for the manuscript. The Referee comments are cited in italics and our responses in regular type while revisions prepared to the manuscript are marked in red.

General and specific comments:

*1) Given the scientific significance and the potentiality of this work, I believe it deserves more visibility. I think the authors are underselling their work. For instance, the title seems to suggest a study with highly technical details which can discourage non-expert readers, whilst could be more general to attract more audience. Consider avoiding the use of KZ in the title, it is just a moving average filter.*

- We thank you for this comment as will allow increasing the visibility of the paper to a broader audience. We took a further step and we will change the title to: "The Use of Hierarchical Clustering for the Design of Optimized Monitoring Networks""

*2) It's not clear to me the average behind figure 9. It shows the correlation map of each grid cell with any other cell? Does it imply an average over R? or it is the time or spatial series correlation being investigated? Please clarify in the text*

- This is a good point, and we have revised the text (below) to try to clarify this issue. Figure 9a and b show the values of 1-R for each grid cell at the point in the analysis where that grid square becomes part of a cluster, therefore no averaging was used. Those grid cells with high values of 1-R thus join clusters at much lower correlation levels than those which have joined clusters at low values of 1-R. The maps show the extent of dissimilarity for the grid cells; higher values show grid cells which are so unlike others that they remain separate from the clusters throughout much of the analysis. In contrast, Figure 9c and d show the clusters which exist for a specific level of 1-R. These show how the methodology may be used to design a monitoring network for a given number of stations (i.e. one station within each of the coloured regions will be sufficient to represent that coloured region, to within the value of 1-R used to generate the clusters).

P14,L5-9 "Figure 9 depicts the resulting mapped 1-R cluster analysis in this area, when each model grid-cell has been treated as a potential monitoring station location. Figure 9 (a and b) show levels the values of 1-R for each grid cell at the point in the analysis where that grid square becomes part of a cluster for NO2 and for SO2, respectively. Those grid cells with high values of 1-R thus join clusters at much lower correlation levels than those which have joined clusters at low values of 1-R. As a result, the maps show the extent of dissimilarity for the grid cells; higher values show grid cells which are so unlike others that they remain separate from the clusters throughout much of the analysis. In contrast, Figure 9 (c and d) show the clusters which exist for a

specific level of 1-R. These show how the methodology may be used to design a monitoring network for a given number of stations (i.e. one station within each of the coloured regions will be sufficient to represent that coloured region, to within the value of 1-R used to generate the clusters). Figure 9c,d shows the spatial distribution of the clusters generated by dissimilarity levels of 0.65 for NO2 and 0.8 for SO2, respectively (these levels were chosen based on the analysis above, where the model was shown to provides reasonable results).

*3) it is not clear how redundancy is defined: Overlapping variance, coefficient of determination above a certain threshold, : : :;*

- Here, redundancy has been defined as the relative dissimilarity level at which a station joins a cluster, with respect to a given metric for clustering. We have modified the text to make this more clear, specifically:

P3 L6-7: "Dissimilarity may thus be used to rank stations in terms of potential redundancy, where we define redundancy as the relative dissimilarity level at which a station joins a cluster. Stations having the lowest levels of dissimilarity may hence be considered sufficiently similar to be considered potentially redundant."

P7 L25-28: "Hierarchical clustering as described above was used to assist in the evaluation of potential monitoring station redundancies (defined as the relative dissimilarity level at which a station joins a cluster), as one of many considerations that could influence decision making on monitoring network design. Having carried out hierarchical clustering using station data, the values of the dissimilarity metric as stations join clusters may be used to define the extent of similarity between stations, as well as a relative ranking of stations based on these similarities."

- We do not define specific levels of dissimilarity since by its nature the analysis provides relative information. We have mentioned that it is up to the decision maker to set the levels of dissimilarity that stations can be considered redundant, as these levels differ substantially depending of the data used (no. of stations, no. of records, species analysed), P8 L8-13: ""Redundancy" with regards to the metrics examined here is thus *relative* to a given chemical species and dataset used for hierarchical clustering. Therefore, we do not propose specific thresholds of the two metrics for determining redundancy. We note also that the results of the analyses for two metrics may be combined – station data that are relatively similar under one metric may be examined for their degree of similarity under another metric. The metric levels at which these combinations are examined are themselves also qualitative, but station time series which are highly similar under multiple metrics are in turn a stronger indication of potential redundancy". Therefore stations will be potentially redundant if stations highly correlate with each other (low 1-R levels) and if the Euclidean distance levels are low. To decide if stations are redundant or not, a level of 1-R and/or Euclidean distance should be set; all the stations clustering under the same cluster at that given level should be under consideration for being removed or moved.

*4) based on this study, can the authors comment on the minimum exposure period (length of time series) for the clustering analysis to be reliable*

- This is a very good question, but difficult to answer with the available data. For example, we seem to have reasonable/useful results for the bimonthly analysis using 5 years of data (30 values), while our hourly analysis of model output includes a year of data (8760 values in the centre of the latter dataset). The ideal answer is "as much data as is available", if, for example, one wants to limit year-to-year variability. This may not always be possible or practical, especially for the deterministic model applications, which can be very computationally expensive. We added a paragraph to Section 7 to point out this issue.

P16,L15: "We note here that the results of analyses of this nature are dependent on the time series data used (including its duration). We have used a 5 year dataset to evaluate bimonthly observation data, and a one-year dataset to evaluate annual data and deterministic model results. Longer time periods may be preferred in future applications to limit the potential impact of year-to-year variability."

*5) page 4 ,line 17. Consider Vardoulakis et al. 2011. Atmospheric Environment 45 (2011) 5069-5078*
- We will revise the original text to accommodate this reference

*6) page 8, line3. Not only 'dissimilarity metrics' but also agglomeration method and definition of correlation coefficient are quite sensitive parameters*
- Thank you for pointing this out, the dissimilarity metric is indeed not the only sensitive parameter. We have revised the original text to accommodate this comment:

P7,L34-35: "1. The ranking of stations is relative and specific to a given chemical species, the corresponding set of station time series, and the parameters used for the hierarchical cluster analysis: the metric of dissimilarity and the method to recalculate the dissimilarity matrix used in the analysis."
P8,L3: "An important corollary to the first point above is that different methods dissimilarity metrics used in hierarchical clustering may result"

*7) Can the Euclidean distance be used to spot systematic detection error?*
- We believe that is possible when comparing with station that have similar features but the user should be knowledgeable of the stations' characteristics, surrounding sources, topography, etc., so the metric value can be analysed properly. For example, stations which are in close spatial proximity yet have substantial differences in Euclidean distance imply systematic detection errors in the monitoring data. We believe though, that 1-R might be a better indicator if the user has no prior or little knowledge of the stations included in the analysis.

*8) page 14 line 16. Can the authors comment on the spatial continuity of the solution? Is it a requirement or the area can contain holes and/or be even detached?*
- There is no inherent requirement on the methodology that a cluster be spatially continuous. An example of this can be seen in Figure 9c, wherein a cluster extending from the centre of the emitting region to the lower-left corner of the map is split into two separate regions (red coloured region, cluster 3). In Figure 9d, the same area does not show the same split. Local knowledge of the emissions sources, as well as analysing Figure 9a and c, help explain these results. The centre of the

grey region (cluster 8) in the lower-left of Figure 9 d, and the corresponding dark yellow (cluster 5) region in Figure 9b, mark the location of a local emissions source, moderate in magnitude relative to the larger sources in the middle of the domain. The clustering thus recognizes the local influence of this emissions source (creating the clustered areas in the two figures). However, at greater distances from this moderate source of emissions, the impact of the major sources in the centre of the domain dominates. The green area (cluster 4) in Figure 9d, and the red areas (cluster 3) in Figure 9c, show that this large source has both a local and long-range influence, which only locally can be overwhelmed by the moderate source for both $SO_2$ and $NO_2$. We note that we are using 1-R in our demonstration here, so the magnitude of the signal of the two chemicals is not being analysed, rather, its time variation. We added text based on the explanation above.

P14,L23: "We note that in some cases a single cluster can be discontinuous, split into more than one area. An example of this can be seen in Figure 9c, where a cluster is split into two separate red coloured regions (cluster 3), whereas Figure 9d does not show the same split. Local knowledge of the emissions sources, as well as analysing Figure 9a and b, help explain these results. The dark yellow region (cluster 5) in Figure 9c and the grey region (cluster 8) in Figure 9d mark the location of a local emissions source, moderate in magnitude relative to the larger sources in the middle of the domain (Oil Sand facility boundaries marked in these Figures). The clustering thus recognizes the local influence of this moderate source of emissions, however, at greater distances from this source, the impact of the larger sources dominates. The red areas (cluster 3) in Figure 9c and the green area (cluster 4) in Figure 9d show that the larger sources have both a local and long-range influence, which only locally can be overwhelmed by the moderate source for both $SO_2$ and $NO_2$. We note that we are using 1-R in this application of the methodology with deterministic model output, so the magnitude of the signal of the two chemicals is not being analysed, rather, its time variation."

*9) Page 15, line37. Solazzo and Galmarini misspelled.*
- We will revise the original text to accommodate this comment.

*10) Page 15, line37. A source of dissimilarity was found to be the reporting time not harmonised across European countries. Data reporting at the beginning or at the end of the hour can make a significant difference*
- We understand the reviewers comment. We will remove the sentence "As mentioned in Sollazzo and Gamarini (2015), the manner in which the data is reported may significantly impact the analysis." as the reference here is not applicable.

I invite the authors to comment on the following:

*I think we are still far away from using clustering for operational use. Clustering is known to provide some qualitative insight, but it is quantitatively weak as it depends on many parameters. Indeed, a fundamental challenge of clustering is the high sensitivity to the options controlling the underlying algorithms, such as the agglomerative method, the distance metric, the number of clusters, and the cut-off distance are aspects that need to be determined case by case. In particular, the cut-off (the threshold similarity above which clusters are to be considered disjointed) determines the dimension of the sub-*

*space of non-redundant information and is decided by visual inspection of the dendrogram. Supervised clustering (e.g. k-Means) initiated with the results of unsupervised clustering might be more robust.*

- The authors agree that there are many controlling factors to the outcome but we see this approach suitable for unsupervised clustering. Even bearing in mind these considerations, the methodology has been shown to provide "real" insights, for example picking out stations which are known to experience significantly different conditions than others (e.g. closer to a major emissions source, or at a remote higher elevation location where no sources usually impact the site). By the same token, the methodology can objectively identify issues of potential concern (such as co-located observation records which are highly dissimilar. We view this methodology can be a starting point for the redundancy decision making and, with the results from this analysis, a more supervised strategy to follow. We also note that while many papers suggest that k-Means give a more robust outcome, if the user has no prior knowledge how the first set of clustering should appear (a requirement for k-means), this will a priori jeopardize a k-Means analysis. We noted that this methodology isolated stations that for technical reasons or specific local characteristics *should* appear different, and therefore have confidence that this methodology is a good starting point for monitoring network analysis.

*The application of associativity analysis for detecting potential redundancy in the context of regulatory air quality monitoring might have some pitfalls (most of which are anyway mentioned by the authors in the text, but I think deserve more words). For example, the potential duplicate of information obeys some policy precautionary principle and might reveal useful in some instances (double checking, reduce missing records, cross validation, etc). Further, redundancy should be determined with some long-term climatology and should also serve future decision making in the sense that what might be redundant based on the past ten years of data might not be in the next ten years. In this sense the adoption of models for future scenarios might help.*

- The reviewer raises good points. We have included the following paragraph into the caveats section of the paper and as an addition to the length of the time series proposed in comment 4)

P16, L15: "Nevertheless, if emissions change in the future, the analysis should be repeated in order to determine whether the pattern of clusters has changed in response to the changes in emissions. Similarly, while long time sets are desired from the standpoint of removing the potential impacts of annual variability in meteorological conditions, if changes in emissions happen frequently, it may argue for yearly rather than multi-year analyses. "

- We agree that redundancy could be determined with some long-term climatology and emission scenarios.

*I think that, more than the estimation of redundancy, the main strengths of the methodology are the potential for classification and the estimation of the area of representativeness (AoR). Indeed I would have framed the whole work in the context of classification.*

- We do agree with the Reviewer that a strength of the work is that it provides an estimation of the AoR of a station (potentially useful for other applications such as data assimilation), but our first goal was to develop a tool for decision makers that only have observational data in hand, and need to assess the potential redundancy of existing stations. Our second goal was to determine the extent to which deterministic models could be used to provide information for future monitoring networks. The clustering maps from the latter application provide AoR information, as well as the potential to account for future changes in emissions (via deterministic model simulations which use projections

of future emissions to determine clusters). To use this work in the context of classification is indeed interesting and worth considering for future manuscripts.

*For example, can the methodology assist in the classification of monitoring station based on area-type or site-type?*
- The authors belive that maps such the ones presented in Figure 9 can be overlayed with information such as population density, emission sources and orography, allowing classification of the monitoring stations.

*Do the authors expect the diurnal signal to be the associated over long distance?*
- We're not completely certain what is meant by "over the long distance" in this context.  The relative impact of the diurnal signal in the time series will depend on the extent to which diurnal variation controls the emissions, transportation and deposition of the given pollutant. As we noted in the paper, we have conducted analyses which suggest that much of the signal which provides information on local conditions (including local emissions sources) resides within the shorter time scales on the order of a day or less. There is of course a diurnal signal that will be present across the larger region, due to the diurnal variation in chemistry associated with the transition between day and night.  Again, our work would suggest that much of the similarity between observation sites resides within that diurnal variation.

*In siting a new station, its area-type can be defined by looking at how the signal of existing stations compares with the signal of the new station?*
- We are assuming that the reviewer is asking whether the analysis can provide ancillary "area of representativeness" information beyond simply describing the shape of regions of equal similarity. Provided the analysis is combined with additional information, some additional information can be derived from figures such as Figure 9.  For example, the $SO_2$ cluster map (9(d)) is similar to a "wind rose" pattern – pie wedge shaped regions extending radially outward from the centre of the emissions region, with a smaller number of smaller and more irregularly shaped clusters in the middle of the emissions region.  $SO_2$ emissions from this region are largely (>90%) from point sources; large stacks which create discrete plumes which are carried downwind and may fumigate to the surface.  For these very discrete sources of $SO_2$, the wedges thus relate to the relative probability that a plume will be carried in a given direction downwind (note that a 1-R metric implies that a plume fumigating downwind over a long distance will have an equal correlation radially outward from the emissions point).  The irregular shaped regions closer to the sources thus represent regions over which very local fumigation under stagnant conditions may take place, "earning" them a separate set of clusters.  This may be contrasted with Figure 9(c), for $NO_2$.  In this region, about 40% of $NO_2$ emissions is from large stacks, while the remainder is from more spatially distributed sources such as the off-road fleet of large diesel vehicles used to haul unprocessed bitumen to processing facilities.  The radial pattern is present, but muted compared to $SO_2$, when the same number of clusters is generated, suggesting the larger impact of the "area" sources for $NO_2$.  So, in this case, the pattern of the clusters is diagnostic of the type of source – large stacks ($SO_2$, a "sharp" radial pattern dominates) versus a combination of large stacks and more distributed area sources ($NO_2$, with broader regions for the radial pattern, and more of the irregular localized clusters).  Note, however, that additional information regarding the source types was required to make this observation, and further work will be needed to determine whether these patterns may be used in the absence of such information to infer area-type.  One can also run the analysis for existing station location and a planned new station to determine how the planned station would compare to an existing network; if existing stations are known to be impacted by a specific source

type, the resulting dissimilarity analysis could be used to determine the "type" of the planned station. Other Graphical Information System datasets could also be overlaid with information such as shown in Figure 9 to help define the relationships between the observed patterns and other geographical information. However, we are reluctant at this stage to convert these observations, which may be specific to the sub-region examined here, to a more general guidance in the text regarding the use of the methodology to provide ancillary area-type data.

*I would invite the authors to add some further considerations about the potentiality of the methodology devised, also in light that some reflections are already part of the paper, for example the clustering of long term signals.*

- The authors will revise the text to add further considerations about the potentiality of the methodology. A paragraph summarizing the potentiality of the methodology was added in the end of Section 4.1, P9, L38:

"In summary, the methodology is able to identify groups of stations which are influenced by common emissions sources (e.g. stations which are influence by oil sands emissions as opposed to stations located elsewhere) when the methodology is applied to hourly and, to some extent, daily time-filtered time series. Stations mainly influenced by seasonality are identified when the methodology is applied to weekly and monthly time-filtered data. The analysis groups stations according to their degree of similarity but does not provide the cause for that degree of similarity. The latter may only be achieved by examination of the data records, and the use of local knowledge of sources and conditions. The level of information about the sources present in the study area will be greater when the results of both metrics are combined, and information about the sources may be inferred from the analysis; for example, stations could be classified as background or industrial impacted if seasonality or hourly data are shown to contain most of the signal."

*Concerning the AoR, the authors (or at least some of them) have already experience with the topic, and I have been surprised that it was not expanded in the text, especially since model results are available. The maps in figure 9 indeed show some AoR! The authors mentioned it at the beginning of page 3 but then drop it. For example, some discussion about AoR would fit nicely in section 4.1. Again, in light of better exploiting the large amount of work done, I would invite the authors to consider adding some further words about the potentiality of the analysis for determining the AoR.*

- The authors thank the Reviewer for the words of encouragement. We admit that Figure 9 was a teaser for future manuscripts, as the authors would like to publish further on this topic. We have added further considerations about the potentiality of the methodology:

P14,L23: "To satisfy different monitoring objectives, stations are placed by both geographical and physical location, with physical location defined by the concept of spatial scale of representativeness, the area where actual pollutant concentrations are reasonably uniform. We note that each of these coloured subregions in which a single station could be placed has a relatively large geographic extent, and, using this metric, do not describe the concentration gradient in the region but could be used as a first guess for areas of representativeness, potentially providing useful input for applications such as data assimilation of air-quality and meteorological observations. Combining spatial distribution of the clusters for 1-R metric with the Euclidean distance will provide further information about the concentration gradients in the area of representativeness."